# Coupling between surface ozone and leaf area index in a chemical transport model: Strength of feedback and implications for ozone air quality and vegetation health

Shan S. Zhou[1], Amos P. K. Tai[1,2], Shihan Sun[1], Mehliyar Sadiq[1], Colette L. Heald[3], Jeffrey A. Geddes[4]

[1] Earth System Science Programme and Graduate Division of Earth and Atmospheric Sciences, Faculty of Science, The Chinese University of Hong Kong, Sha Tin, Hong Kong

[2] Partner State Key Laboratory of Agrobiotechnology and Institute of Environment, Energy and Sustainability, The Chinese University of Hong Kong, Sha Tin, Hong Kong

[3] Department of Civil and Environmental Engineering and Department of Earth, Atmospheric and Planetary Sciences, Massachusetts Institute of Technology, Cambridge, USA

[4] Department of Earth and Environment, Boston University, Boston, USA

*Correspondence to*: Amos P. K. Tai (amostai@cuhk.edu.hk)

**Abstract.** Tropospheric ozone is an air pollutant with substantial harm on vegetation, and is also strongly dependent on various vegetation-mediated processes. The interdependence between ozone and vegetation may constitute feedback mechanisms that can alter ozone concentration itself but have not been considered in most studies to date. In this study we examine the importance of dynamic coupling between surface ozone and leaf area index (LAI) in shaping ozone air quality and vegetation. We first implement an empirical scheme for ozone damage on vegetation in the Community Land Model (CLM), and simulate the steady-state responses of LAI to long-term exposure to a range of prescribed ozone levels (from 0 ppb to 100 ppb). We find that most plant functional types suffer a substantial decline in LAI as ozone level increases. Based on the CLM-simulated results, we develop and implement in the GEOS-Chem chemical transport model a parameterization that computes fractional changes in monthly LAI as a function of local mean ozone levels. By forcing LAI to respond to ozone concentrations on a monthly timescale, the model simulates ozone-LAI coupling dynamically via biogeochemical processes including biogenic volatile organic compound (VOC) emissions and dry deposition, without the complication from meteorological changes. We find that ozone-induced damage on LAI can lead to changes in ozone concentrations by –1.8 ppb to +3 ppb in boreal summer, with a corresponding ozone feedback factor of –0.1 to +0.6 that represents an overall self-amplifying effect from ozone-LAI coupling. Substantially higher simulated ozone due to strong positive feedbacks is found in most tropical forests, mainly due to the ozone-induced reductions in LAI and dry deposition velocity, whereas reduced isoprene emission plays a lesser role in these low-$NO_x$ environments. In high-$NO_x$ regions such as eastern US, Europe and China, however, the feedback effect is much weaker and even negative in some regions, reflecting the compensating effects of reduced dry deposition and reduced isoprene emission (which reduces ozone in high-$NO_x$ environments). In remote, low-LAI regions including most of the Southern Hemisphere, the ozone feedback is generally slightly negative due to the reduced transport of $NO_x$-VOC reaction

products that serve as NO$_x$ reservoirs. This study represents the first step to account for dynamic ozone-vegetation coupling in a chemical transport model with ramifications for a more realistic joint assessment of ozone air quality and ecosystem health.

## 1 Introduction

Tropospheric ozone (O$_3$) is an important greenhouse gas with an estimated radiative forcing of 0.40±0.20 W m$^{-2}$ [*IPCC*, 2013]. It is also an important air pollutant shown to have harmful effects on both human health and vegetation, including crops [*Anenberg et al.*, 2010; *Ainsworth et al.*, 2012]. Tropospheric ozone is primarily produced from the photochemical oxidation of various precursor species including carbon monoxide (CO), methane (CH$_4$) and volatile organic compounds (VOCs) by hydroxyl radicals (OH) in the presence of nitrogen oxides (NO$_x \equiv$ NO + NO$_2$). Most of the precursors have large anthropogenic sources from industrial and agricultural activities, and tropospheric ozone concentrations have been increasing since the industrial revolution. The earliest surface ozone observations recorded at L'Observatoire de Montsouris near Paris showed an annual mean ozone concentration of 11 ppb for the 1876-1910 period in Europe [*Volz and Kley*, 1988]. Ozone concentrations displayed a significant upward trend at northern midlatitudes during 1970s-1980s, and then a flattening or even declining trend depending on the region in the last two decades [*Oltmans et al.*, 2013]. As anthropogenic emissions are expected to decrease in many countries due to more stringent regulation [van Vuuren et al., 2011], other factors such as climate, land surface and vegetation changes will likely have increasingly important roles shaping future ozone levels [e.g., Tai et al., 2013; Wong et al., 2017]. In this study, we in particular examine the possible roles of two-way interactions between ozone and vegetation in modulating surface ozone air quality using a coupled land-atmosphere modeling framework.

Vegetation can influence both the sources and sinks of tropospheric ozone. Globally, precursors from natural sources play an important role in ozone formation. They include such gases as NO$_x$, CH$_4$ and various non-methane VOCs (NMVOCs) emitted by land vegetation and soil microbes. Isoprene (C$_5$H$_8$), which is the most abundant biogenic NMVOCs species emitted by vegetation [*Guenther et al.*, 2006], is a major ozone precursor in high-NO$_x$ environments, but can consume ozone by direct ozonolysis or reduce ozone by sequestering NO$_x$ via the formation of isoprene nitrate and peroxyacetyl nitrate (PAN) in low-NO$_x$ environments [*Horowitz et al.*, 2007; *Hollaway et al.*, 2017]. On the other hand, the major sinks for surface ozone include in-situ chemical loss mainly via photolysis and the subsequent reaction of singlet oxygen atom O($^1$D) with water vapor (H$_2$O), and the dry deposition of ozone onto vegetated surfaces [*Wang et al.*, 1998; *Wild*, 2007]. Leaf stomatal uptake of ozone, in particular, represents 40–60% of the total dry-depositional sink [*Fowler et al.*, 2009]. Vegetation also controls transpiration, which modulates boundary-layer mixing, temperature, water vapor content, and thus the production, dilution and loss of ozone. Therefore, via biogenic VOC emissions, dry deposition and transpiration, vegetation can substantially influence surface ozone concentrations.

Surface ozone can in turn influence vegetation. The stomatal uptake of ozone has been shown to damage plants and reduce photosynthetic CO$_2$ assimilation at the leaf level, which may in turn reduce leaf area index (LAI), gross primary productivity (GPP), and crop yield [*Karnosky et al.*, 2007; *Ainsworth et al.*, 2012]. *Yue and Unger* [2014] developed a terrestrial

ecosystem model to assess the damage of surface ozone on gross primary productivity (GPP) throughout the US, and found that GPP is reduced by 4-8% on average in the eastern US in the growing season due to exposure to year-2005 ozone levels. Another study also found that global GPP and transpiration are reduced by 11% and 2.2%, respectively, under exposure to present-day ozone concentrations, with the greatest damage (20–25% for GPP, 15–20% for transpiration) happening at

northern midlatitudes [*Lombardozzi et al.*, 2015]. The ozone-induced decrease in transpiration has been shown to enhance regional temperature by up to 3°C and reduce precipitation by up to 2 mm d$^{-1}$ in summertime central US [*Li et al.*, 2016]. Differential abilities of plant species to tolerate ozone, when integrated over space and time, can also cause the long-term shifts in species richness and ecosystem composition [e.g., *Fuhrer et al.*, 2016]. As vegetation variables such as stomatal resistance, LAI, and plant functional type (PFT) distribution all play important roles shaping surface ozone, dynamic changes in these

variables following ozone damage may thus induce a cascade of feedbacks that ultimately affect ozone itself. The impact of such ozone-vegetation coupling on ozone air quality has only recently been examined by *Sadiq et al.* [2017], who found that by implementing synchronous ozone-vegetation coupling in the Community Earth System Model (CESM), simulated present-day surface ozone concentrations can be higher by 4–6 ppb over North America, Europe and China. Roughly half of such an enhancement is caused by reduced ozone dry deposition following increased stomatal resistance, and the rest mostly arises

from reduced transpiration that leads to higher vegetation temperature and thus isoprene emission. They suggested that a major challenge in diagnosing the various feedback pathways is the high uncertainty associated with the temperature and precipitation responses to transpiration changes. This complication from meteorological feedbacks could mask the relative importance of individual vegetation variables (e.g., LAI) in contributing to the overall coupling effect, rendering attribution more difficult.

LAI is a ubiquitously important land surface parameter driving atmospheric chemistry and hydroclimate in many

models [e.g., *Wong et al.*, 2018]. Previous modeling studies of ozone damage on vegetation usually used prescribed LAI and other structural variables such as canopy height derived from satellites and land surveys as fixed parameters in their vegetation simulations [e.g., *Sitch et al.*, 2007; *Lombardozzi et al.*, 2015]. Some studies have considered the long-term decline in growth and LAI following ozone damage on GPP, which further limits GPP itself [*Yue and Unger*, 2015] and can lead to a biogeochemical feedback effect on air quality due to the importance of LAI in modulating surface ozone and hydrometeorology

[*Sadiq et al.*, 2017]. Observation-based, prescribed LAI may be adequate for present-day or short-term simulations of vegetation and climate, but with ozone-vegetation interaction as well as other drivers such as warming and rising CO$_2$, historical and future foliage properties may be significantly different from those in the present day. Prognostic LAI simulated dynamically by biogeochemical models may therefore be required to enable more realistic simulations especially for multidecadal and century-long historical simulations or future projections under rather different climate scenarios.

Furthermore, while ozone concentration generally responds and equilibrates with any changes in the terrestrial boundary conditions over relatively short timescales (hours to weeks), the responses of vegetation to ozone exposure are usually slower and may require months to years to fully take effect due to the cumulative nature of ozone damage [*Lombardozzi et al.*, 2012]. It is essential to examine different timescales of ozone-vegetation coupling, and decide accordingly the most suitable and computationally efficient model coupling approach to adequately capture the interactive effects.

In this study, we focus on the relationship between ozone and LAI, and assess its relative importance in the overall ozone-vegetation coupling effect. We use a standalone land surface model (LSM) with active biogeochemical cycles driven by prescribed meteorological inputs and ozone concentrations to examine the long-term evolution of LAI in response to different levels of ozone exposure. Based on the simulated ozone-LAI relationships, we develop a simplified parameterization scheme for synchronous coupling between ozone and LAI on a monthly timescale for computationally efficient use in air quality assessment by a chemical transport model (CTM). We also investigate the effect of asynchronous coupling by performing a series of offline-coupled LSM-CTM model experiments. By comparing simulated ozone concentrations from CTM simulations with vs. without ozone-induced damage on LAI, we quantify the "ozone feedback" that results from ozone-vegetation coupling, and examine the possible pathways contributing to the feedback. This effort not only allows ozone-vegetation coupling to be considered dynamically within an atmospheric model without the complication from meteorological changes and feedbacks, but also renders the incorporation of ozone-induced biogeochemical feedbacks and air quality-ecosystem coevolution more computationally affordable in regional climate and air quality models.

## 2 Model description and simulations

### 2.1 Basic description for Community Land Model

In this study, we simulate ozone damage on vegetation using the Community Land Model (CLM) version 4.5, embedded within the Community Earth System Model (CESM) version 1.2.2 forced by prescribed atmospheric data from Climate Research Unit (CRU)–National Centers for Environmental Prediction (NCEP) at a resolution of 1.9° latitude by 2.5° longitude. This version not only updates important canopy processes including canopy radiation and upscaling of leaf processes from previous versions, but also improves the stability of the iterative solution in the computation of photosynthesis and stomatal conductance [*Sun et al.*, 2012]. We use the "BGC mode" with active biogeochemistry [*Oleson et al.*, 2013], which dynamically simulates ecosystem structural variables (LAI and canopy height) based on post-photosynthesis carbon allocation. When evaluated against regional observations, this version of CLM typically captures the spatial variability of gross primary production (GPP) and LAI well, albeit with different signs and degrees of region-specific biases including a general overestimation in both variables in humid, highly productive regions [e.g., *Wang et al.*, 2015; *Sakalli et al.*, 2017].

CLM4.5 uses the Ball-Berry stomatal conductance model [*Ball et al.*, 1987] described by *Collatz et al.* [1991] to simulate leaf stomatal conductance ($g_s$) as

$$g_s = \frac{1}{r_s} = m \frac{A_n}{c_s} h_s P_{\text{atm}} + b \tag{1}$$

where $r_s$ is the leaf stomatal resistance (s m$^2$ μmol$^{-1}$), $m$ is a PFT-dependent parameter ($m = 9$ for C$_3$ plants and $m = 4$ for C$_4$ plants), $A_n$ is the leaf net photosynthesis rate (μmol CO$_2$ m$^{-2}$ s$^{-1}$), $c_s$ is the CO$_2$ partial pressure at the leaf surface (Pa), $h_s = e_s/e_i$ is the leaf surface fractional humidity with $e_s$ being the vapor pressure (Pa) at the leaf surface and $e_i$ being the saturation vapor

pressure (Pa) at leaf temperature, $P_{atm}$ is the atmospheric pressure (Pa), and $b$ is the minimum stomatal conductance (μmol m$^{-2}$ s$^{-1}$) ($b$ = 10000 for C$_3$ plants and $b$ = 40000 for C$_4$ plants) when there is no net photosynthesis.

The rate of net photosynthesis, $A_n$, is computed based on the Farquhar model [*Farquhar et al.*, 1980] for C$_3$ plants, and the photosynthesis scheme of *Collatz et al.* [1992] for C$_4$ plants. Overall, $A_n$ (μmol CO$_2$ m$^{-2}$ s$^{-1}$) is represented by

$$A_n = \min(A_c, A_j, A_p) - R_d \tag{2}$$

where $A_c$ is the Rubisco (RuBP carboxylase/oxygenase)-limited photosynthesis rate, $A_j$ is the RuBP-limited photosynthesis rate, $A_p$ is the product-limited photosynthesis rate, and $R_d$ is the dark respiration rate, all in the same unit as $A_n$. The photosynthesis rate is dependent on intercellular CO$_2$ concentration ($c_i$), which is in turn dependent on $g_s$, $c_s$, and ambient CO$_2$ concentration ($c_a$) through the diffusive flux equations. $A_n$ and $g_s$ are therefore strongly coupled and at every model time step a unique solution for $A_n$ and $g_s$ is found by numerical iterations until $c_i$ converges.

## 2.2 Scheme for ozone damage on vegetation

In the default configuration, CLM calculates stomatal conductance, which controls both water and carbon fluxes, tightly coupled to photosynthesis as mentioned above. Ozone-mediated impacts on vegetation are not included. Several land models have incorporated ozone damage by directly modifying photosynthesis using an empirical ozone flux-based factor, which in turn affects stomatal conductance [*Sitch et al.*, 2007; *Yue and Unger*, 2014]. *Lombardozzi et al.* [2012] showed that modifying photosynthesis and stomatal conductance independently using different ozone impact factors can improve model simulations of vegetation responses to ozone exposure. In this study, we implement the ozone damage scheme of *Lombardozzi et al.* [2015], which modifies the initial net photosynthesis rate ($A_n$) and stomatal conductance ($g_s$) calculated by the Farquhar-Ball-Berry model (described above) independently using two ozone damage factors, $F_p$ and $F_c$, which are multiplied to $A_n$ and $g_s$, respectively. These two factors are calculated from the cumulative uptake of ozone (CUO), which integrates the ozone flux through leaf stomata over the growing season or leaf lifetime:

$$CUO = 10^{-6} \sum \frac{[O_3]}{k_{O_3} r_s + r_b + r_a} \Delta t \tag{3}$$

where [O$_3$] is the surface ozone concentration (nmol m$^{-3}$), $k_{O_3}$ = 1.67 is the ratio of leaf resistance to ozone to leaf resistance to water, $r_s$ here is the leaf-level stomatal resistance, $r_b$ is the leaf boundary layer resistance, and $r_a$ is the aerodynamic resistance between the leaf and the reference level, and $\Delta t$ = 30 min is the given model time step. CUO is only accumulated when LAI is above a minimum value of 0.5 and ozone flux is larger than a critical threshold of 0.8 nmol O$_3$ m$^{-2}$ s$^{-1}$ to account for the compensating ability of plants to detoxify ozone [*Lombardozzi et al.*, 2012; 2015]. The two ozone damage factors are calculated as a linear function of CUO as:

$$F_p = a_p \times CUO + b_p \tag{4}$$

$$F_c = a_c \times CUO + b_c \tag{5}$$

where $a_p$, $b_p$, $a_c$ and $b_c$ are empirical slopes and intercepts (Table 1). There are 15 PFTs (*Supplement* Table S1) plus bare ground in the vegetation composition of CLM4.5 [*Oleson et al.*, 2013], but as Table 1 shows, the experimental effects of ozone differ

among three more general plant groups: broadleaf trees, needleleaf trees, and grasses and crops. We therefore lump CLM4.5 PFTs into the three plant groups: "broadleaf" to include all broadleaf tree and broadleaf shrub PFTs, "needleleaf" to include all needleleaf tree and shrub PFTs, and "grasses and crops" to include $C_3$ and $C_4$ grasses and $C_3$ unmanaged rainfed crops.

**Table 1: Slopes (per mmol m$^{-2}$) and intercepts (unitless) used in Eq. (4) and Eq. (5) to relate cumulative uptake of ozone (CUO) to the ozone damage factors applied to the net photosynthesis rate and stomatal conductance, following *Lombardozzi et al.* [2015]. Values for "average" sensitivity (as opposed to "high" and "low" sensitivity) are used in this study.**

|  | Photosynthesis | | Conductance | |
| --- | --- | --- | --- | --- |
|  | Slope, $a_p$ | Intercept, $b_p$ | Slope, $a_c$ | Intercept, $b_c$ |
| Broadleaf | 0 | 0.8752 | 0 | 0.9125 |
| Needleleaf | 0 | 0.839 | 0.0048 | 0.7823 |
| Grasses and crops | -0.0009 | 0.8021 | 0 | 0.7511 |

**2.3 Description for GEOS-Chem chemical transport model**

We use the GEOS-Chem global 3-D chemical transport model version 10-01 (geos-chem.org) with fully coupled $O_3$-$NO_x$-hydrocarbon-aerosol chemical mechanism for atmospheric chemistry simulations, driven by assimilated meteorological fields from the Goddard Earth Observing System (GEOS-5) produced by the NASA Global Modeling and Assimilation Office

(GMAO), with a horizontal resolution of 2° latitude by 2.5° longitude and 47 vertical layers. GEOS-Chem has been extensively used in ozone simulations and evaluated with in situ and satellite observations in previous studies, both on a global scale [e.g., *Liu et al.*, 2006; *Zhang et al.*, 2010] and a regional scale [e.g., *Wang et al.*, 2009; *Wang et al.*, 2011]. In general, GEOS-Chem underestimates tropospheric ozone in the tropics but overestimates it in the northern subtropics and southern midlatitudes [*Zhang et al.*, 2010]. For regional surface ozone, the model has small systematic biases overall in the US and China, but has a

tendency to overestimate summertime concentrations in the eastern US and certain sites in China [*Wang et al.*, 2009; *Wang et al.*, 2011; *Zhang et al.*, 2011]. Anthropogenic emissions of $NO_x$, CO, sulfur dioxide ($SO_2$) and ammonia ($NH_3$) are from the EDGAR v4.2 (Emissions Database for Global Atmospheric Research) seasonal global base emission inventory for the years 1970–2008. Anthropogenic VOC emissions are from the RETRO (REanalysis of the TROpospheric chemical composition) monthly global inventory for year 2000. Biomass burning emissions are from the year-specific GFED4 (Global Fire Emissions

Dataset) dataset.

In GEOS-Chem, LAI affects surface ozone mainly through three channels: biogenic VOC emissions, dry deposition, and soil $NO_x$ emission. Biogenic emissions are calculated within GEOS-Chem by the Model of Emissions of Gasses and Aerosols from Nature (MEGAN v2.1) [*Guenther et al.*, 2012]. The emission of a given VOC species is based on a baseline

emission factor modulated by a series of activity factors accounting for variations in light, temperature, leaf age, soil moisture, LAI, and $CO_2$ inhibition. Dry deposition is computed by the resistance scheme of *Wesely* [1989], whereby dry deposition velocity is the inverse of aerodynamic resistance ($R_a$), sublayer resistance ($R_b$) and bulk surface resistance ($R_c$) added in series. The term $R_c$ accounts for a combination of resistances from vegetation (including stomatal resistance), lower canopy and ground, which have specific values for 11 different land types. *Wong et al.* [2018] extensively evaluated the LAI dependence of both MEGAN biogenic emissions and dry deposition in GEOS-Chem. Soil $NO_x$ emission is based on the scheme of *Hudman et al.* [2012], and further modulated by a reduction factor to account for within-canopy $NO_x$ deposition [*Jacob and Bakwin*, 1991].

Different modules in GEOS-Chem and CLM4.5 have traditionally used a variety of land type and PFT classification schemes. To harmonize between them, we use the land cover harmonization module recently developed by *Geddes et al.* [2016], which classifies vegetation into the same 15 PFTs as CLM4.5 (Table S1) in GEOS-Chem. Emission factors and fractional coverages for those PFTs related to biogenic VOC emissions are mapped and regridded at model initialization. The 15 PFTs are also remapped to the 24 biomes [*Steinkamp and Lawrence*, 2011] for the soil $NO_x$ module according to their types and locations, and to the 11 land types used in the dry deposition module. Original monthly mean LAI input in GEOS-Chem derived from the Moderate Resolution Imaging Spectroradiometer (MODIS) satellite instrument at a grid-level resolution of 0.5°×0.5° is replaced by monthly PFT-level LAI from default CLM land surface data of the present day (year 2000), which is in turn derived from the grid-level MODIS LAI using the deaggregation methods of *Lawrence and Chase* [2007].

## 2.4 Model experiments to determine ozone-LAI relationship

We implement the ozone damage scheme described above into CLM4.5-BGC and conduct 11 simulations under prescribed constant ozone levels from 0 ppb to 100 ppb with an interval of 10 ppb, where the simulation with 0 ppb ozone is treated as the control case (CTR) without ozone damage on vegetation. All simulations are run with initial conditions for year 2000 (which have themselves been obtained from a spin-up simulation starting from no vegetation for more than 1000 years, driven by prescribed year-2000 meteorology) for a total of 80 years. We find that vegetation structure reaches a steady state with no further temporal trends in the monthly mean values roughly after 20-40 years of simulations depending on the prescribed ozone level and PFT. Monthly mean PFT-level one-sided exposed LAI averaged over the last 15 simulation years is extracted as the steady-state solution to be compared with the control case. Aggregate, grid-level LAI, i.e., the fraction-weighted sum over all PFTs, is also calculated.

To quantify ozone damage on vegetation structure, we define a PFT-level ozone impact factor, $\gamma$, to represent the relative change of monthly mean LAI between the case with a given ozone concentration and the control case. The PFT-level $\gamma$ factors are directly calculated as:

$$\gamma_{\text{raw}} = \frac{\text{LAI}_{[O_3]}}{\text{LAI}_{\text{CTR}}} \tag{7}$$

where $\gamma_{raw}$ is a spatially and monthly varying ozone impact factor dependent on ozone concentration directly from the CLM simulations, $LAI_{[O3]}$ is the simulated steady-state monthly LAI at a given ozone concentration, and $LAI_{CTR}$ is the original monthly LAI in the control case with no ozone damage. We find that monthly $\gamma_{raw}$ (both PFT-level and grid-level) for a given location generally decreases as ozone concentration increases, but its decrease per unit ozone increase becomes smaller at higher ozone concentrations (Fig. 1) because of the progressive closure of stomata as represented by the ozone damage scheme. When the stomatal conductance is small enough to limit ozone flux below the critical threshold, no additional damage will be caused by ozone. This restrictive effect from attenuated stomatal conductance on ozone flux prevents LAI from declining infinitely. Thus, above a certain high-enough ozone concentration, $\gamma_{raw}$ generally levels off due to a relatively steady $LAI_{[O3]}$ in the majority of land grid cells worldwide. In some places especially in grasslands and semiarid regions, however, $\gamma_{raw}$ increases with ozone level but its increase per unit ozone increase also declines and then levels off at higher ozone concentrations.

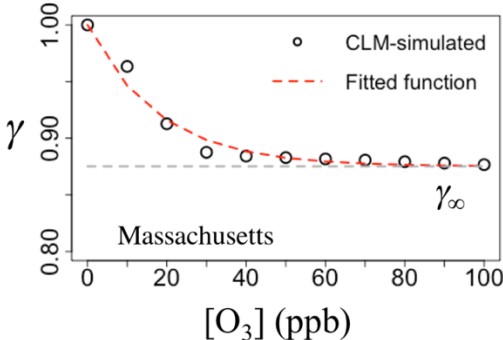

**Figure 1: Relationship between the ozone impact factor, $\gamma$, for relative LAI changes in Eq. (7) and surface ozone concentration in summer (JJA), using the grid cell covering Massachusetts as an example. Black circles refer to $\gamma_{raw}$ directly calculated from CLM simulated results, and the red dashed line refers to $\gamma_{opt}$ in Eq. (8) that is obtained by best-fitting.**

For both kinds of ozone-LAI relationship, we can best-fit an exponential-like function to the values of CLM-simulated $\gamma_{raw}$ with two optimization parameters to obtain an optimized ozone impact factor, $\gamma_{opt}$, for each model grid cell, month and PFT:

$$\gamma_{opt} = \gamma_\infty + (1 - \gamma_\infty)e^{-k[O_3]} \tag{8}$$

where one of the parameter to optimize, $\gamma_\infty$, is essentially the "saturated" relative change in LAI at very high ozone concentrations, and $k$ is the exponential decay factor indicating how "quickly" $\gamma$ evolves with increasing ozone concentration. We find that for 90% of the grid cells/PFTs, $\gamma_\infty$ is between about 0.3 and 1.5; for 50% of the grid cells/PFTs, $\gamma_\infty$ is between about 0.75 and 0.95. To exclude outlying conditions where model-simulated LAI is unrealistically too sensitive to ozone

(mostly in low-LAI regions at the peripheries of major forests and grasslands), we constrain the optimized $\gamma_\infty$ to be between 0.3 and 1.5. We also smooth the fitted values by replacing them with mean values of their surrounding grid cells if the ratio of fitted sum of squares to total sum of squares in that grid cell falls below 0.25 (i.e., if the fitted curve explains less than 25% of the variability of the simulated results). The fitting is done for both PFT-level and grid-level data, and for the vast majority of

PFTs and grid cells the fitted to total sum of squares ratio is above 0.25, demonstrating the robustness of Eq. (8) as the fitting and parameterization function to be implemented in GESO-Chem (or any other CTM or climate model). The maps for fitted $\gamma_\infty$ and $k$ for different PFTs are shown in Figs. S1-2 in the supplement. The global median values of annually averaged $(\gamma_\infty - 1) \times 100\%$ for different PFTs range between –19% (for needleleaf evergreen boreal trees) to +5.0% (for broadleaf deciduous temperate shrubs), and are negative for most PFTs, indicating a general decline of LAI at very high ozone levels. When we

apply some generic ambient ozone level (e.g., 30 ppb) and an elevated (+50%) ozone level to Eq. (8), the percentage changes in LAI as ozone increases from ambient to elevated level range between about –20% and +3% (with the bottom and top 2.5% of grid cells/PFTs trimmed). These modeled LAI changes generally fall within the empirical uncertainty bounds found by previous ozone-elevation experiments for a few species of trees and crops [e.g., *Karnosky et al.*, 2005; *Dermody et al.*, 2008; *Feng et al.*, 2008].

## 2.5 GEOS-Chem experiments with ozone-vegetation coupling

We implement the parameterization equation, Eq. (8) in GEOS-Chem, and conduct three GEOS-Chem experiments (Table 2): 1) *[Intact LAI]*, a control case with monthly prescribed intact potential LAI that is unaffected by ozone; 2) *[Affected LAI]*, an experimental case with LAI being updated continuously and evolving with ozone concentration; and 3)

*[Intact_NoAnth]*, a case with intact potential LAI but without anthropogenic emissions, to examine the strength of ozone feedback (see Sect. 6). We run the above three cases first using 2009 to 2012 meteorology as spin-up and then loop over 2012 meteorology for three simulation years to reach a quasi-steady state for ozone air quality representative of current-day conditions. The detailed implementation algorithm for each simulation is discussed below.

**Table 2: GEOS-Chem experiments to investigate the effect of synchronous ozone-vegetation coupling.**

| Name | Description |
| --- | --- |
| *[Intact LAI]* | GEOS-Chem simulation run with monthly intact potential LAI unaffected by ozone |
| *[Affected LAI]* | GEOS-Chem simulation run with continuously-updated LAI affected by ozone month by month |

| *[Intact_NoAnth]* | GEOS-Chem simulation run with monthly intact potential LAI unaffected by ozone and with no anthropogenic emissions of ozone precursors |
| --- | --- |

For the *[Intact LAI]* case, we first derive an intact, potential LAI that should represent the maximum LAI possible if there is no ozone damage in reality, which is taken as the baseline case for investigating the effect of ozone-LAI coupling. This is necessary because the present-day satellite-derived "observed" LAI is supposedly already the outcome of long-term ozone-vegetation interactions, and should not be used as the baseline. Instead, the potential LAI is used as the initial condition to drive the GEOS-Chem simulations. The potential LAI is derived using the current LAI and optimal $\gamma_{opt}$ factor:

$$LAI_{pot} = \frac{LAI_{MOD}}{\gamma_{opt}} \qquad (9)$$

$LAI_{MOD}$ here is the monthly mean PFT-level LAI from default CLM land cover for the present day originally derived from grid-level MODIS LAI. The optimal $\gamma_{opt}$ is calculated using Eq. (8) from the monthly mean ozone concentrations averaged over year 2005-2008 to represent a present-day norm on which our model experiments are based.

For the *[Affected LAI]* case, PFT-level LAI input of each simulated month is adjusted and evolves dynamically with $\gamma_{opt}$ based on the ozone concentration of the previous month. Specifically, monthly mean ozone concentration of the previous month is read in the first time step of the current simulation month and used to calculate $\gamma_{opt}$ for every PFT in each grid cell, which is then multiplied to the intact potential LAI to derive a new ozone-affected LAI ($LAI_{O3}$) input for the current simulation month:

$$LAI_{O_3} = \gamma_{opt}\, LAI_{pot} \qquad (10)$$

This manner of implementation essentially enables dynamic coevolution of LAI and ozone month by month, and assumes that LAI responds to fluctuations in ozone levels on a monthly timescale.

Such a monthly to intraseasonal timescale for ozone-LAI coupling may not be long enough for plants to fully respond to large fluctuations of ozone concentration. As described in Sect. 2.4, when ozone is incidentally increased from zero to a prescribed level, LAI typically responds and stabilizes over about two decades. As such, is it justifiable to use the relationship between monthly mean ozone and LAI, which have reached a long-term quasi-steady state in CLM, as the basis for parameterization? We first note that, in reality, observed ozone and LAI have likely been through an extended period of coevolution and are likely coupled in a manner resembling in a quasi-steady state, albeit with seasonal fluctuations and some long-term trends. Moreover, the month-to-month variations in ozone concentration are typically much smaller than an incidental jump between zero and a prescribed level. As shown by Fig. 1, the strongest response of LAI to ozone happens at low ozone levels, while at and above ambient levels (e.g., > 30 ppb), LAI responses to any ozone variations also become progressively small. We thus assume that the "steady-state response" for ozone-LAI relationship is reasonably robust to represent short-term responses of LAI superimposed on the long-term, ozone-induced decline in seasonally varying LAI. This

assumption is further tested by driving CLM with hourly varying GEOS-Chem ozone fields in an asynchronous coupling experiment until LAI reaches a quasi-steady state, which will be discussed in Sect. 5.

All simulations use the same prescribed meteorological fields and PFT fractional coverage. Output variables for boreal summer (June, July and August, or JJA), boreal winter (December, January and February, or DJF), and the whole year are extracted for analysis and comparison; in the rest of this paper we will focus on boreal summer results at quasi-steady state, because this is when high ozone concentrations overlap to the greatest extent with the growing season of the majority of land plants in major populated regions at midlatitudes. Equivalent results for boreal winter and the whole year are included in the supplement and discussed briefly in the main text.

## 3 Impact of ozone exposure on leaf area index

We first calculate 2012 JJA mean total LAI by summing over all PFT-level LAI values weighted by the respective PFT fractional coverage, and compare the grid-level $LAI_{O3}$ with $LAI_{pot}$ to examine the impacts of synchronous ozone-vegetation coupling on LAI. Figure 2a shows the distribution of summertime ozone-affected LAI with a maximum value of 5.5 in Amazonia. The spatial pattern of intact, potential LAI (*Supplement* Fig. S3a) is very similar to that of affected LAI, but the magnitude is higher almost everywhere globally. The differences between these two sets of summertime LAI (*[Affected LAI] – [Intact LAI]*) are shown in Fig. 2b (the corresponding percentage changes are shown in Fig. S3b). Due to synchronous ozone-vegetation coupling on a monthly timescale, LAI values generally decline in most of the vegetated regions, and LAI experiences the greatest reduction of up to 2.6 in heavily forested regions including equatorial Asia and southeastern China. There are a few grid cells showing an opposite effect with a slight LAI increase located at the border of vegetated areas, and the possible reasons include "self-healing" effect of vegetation under moderate ozone exposure (e.g., higher water-use efficiency and increased carbon allocation to leaves) to compensate for the photosynthetic damage [*Sadiq et al.*, 2017], as well as numerical outliers due to LAI fluctuations in low-LAI regions. Since the LAI increase in those grid cells is relatively small compared with the magnitude of LAI reduction, and almost always occurs in low-LAI or marginal areas between vegetated and non-vegetated regions, the overall impacts of those grid cells are deemed negligible.

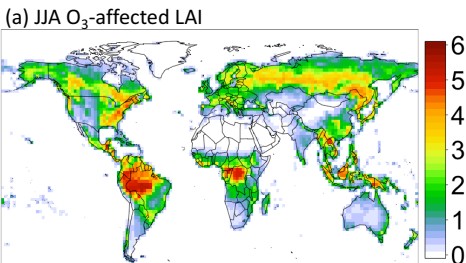

(a) JJA O$_3$-affected LAI

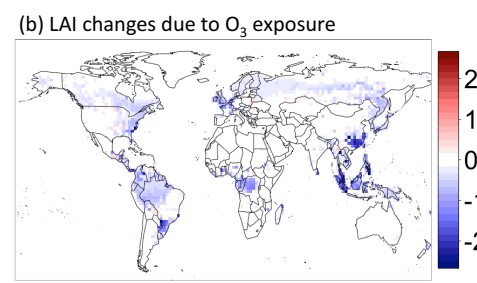

(b) LAI changes due to O$_3$ exposure

**Figure 2: (a) Simulated leaf area index (LAI) affected by long-term ozone exposure in summer (JJA mean) in GEOS-Chem, and (b) differences between ozone-affected LAI in (a) and intact, potential LAI unaffected by ozone, i.e., *[Affected LAI] – [Intact LAI]*.**

We also plot the PFT-level LAI changes between *[Affected LAI]* and *[Intact LAI]* case (Fig. S4), and find that total LAI changes in different places are mainly caused by the dominant local PFTs: in tropical regions such as Amazonia, part of central Africa (mostly in Gabon and Congo) and equatorial Asia, the LAI changes are mainly ascribed to tropical broadleaf evergreen trees; whereas in high-latitude regions, boreal needleleaf evergreen trees play the dominant role in the LAI changes. We also find a large contribution from C3 and C4 grasses and rainfed crops in total LAI reduction in several subtropical and tropical regions such as southeastern China, southern Brazil, part of western Europe, and maritime Southeast Asia. This is likely because of the higher sensitivity of grasses and crops to ozone exposure compared with other plant groups in this ozone damage scheme (see Table 1), and the general overestimation of grass and crop LAI in CLM, which is also documented in other studies [*Chen et al.*, 2015; *Williams et al.*, 2016]. The over-representation of grasses and crops in these subtropical and tropical regions in the CLM world may in turn lead to possible high biases in the simulated ozone damage on LAI therein.

## 4 Impact of synchronous ozone-vegetation coupling on surface ozone

Summertime (JJA) global ozone concentrations and changes due to synchronous ozone-LAI coupling are shown in Fig. 3. Figure 3a shows the JJA mean surface ozone field in the *[Affected LAI]* case, with a global average ozone concentration of 28 ppb and highest value of 75 ppb in central Africa. Figure 3b shows the differences in ozone concentration between the *[Affected LAI]* and *[Intact LAI]* case (see percentage changes in Fig. S5). With LAI being dynamically influenced by ozone in the *[Affected LAI]* case, simulated ozone concentration is generally higher in most vegetated areas such as the tropics, eastern North America, and southern China by up to 3 ppb (~10%), reflecting a significant positive feedback arising from ozone-LAI coupling. The spatial patterns of both the absolute and percentage changes in simulated ozone on the continents generally match that of LAI changes due to ozone exposure (Fig. 2b), whereas ozone concentrations over the oceanic and desert areas also increase, which is likely due to the remote transport of ozone and $NO_x$ reservoir species from high-ozone areas. We also find a slight ozone reduction in north China within 1.8 ppb and in central North America within 0.8 ppb.

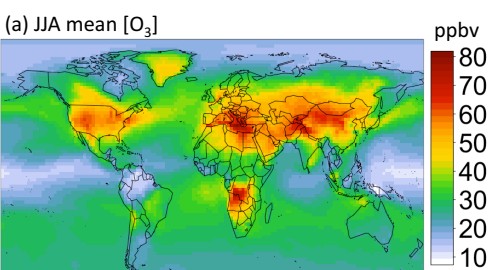
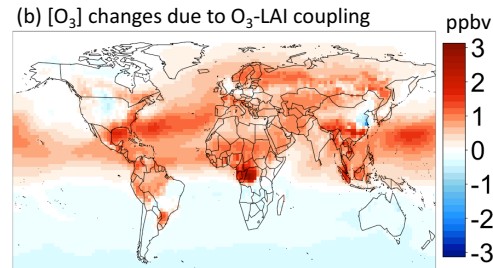

**Figure 3: (a) Surface ozone concentration with ozone-affected leaf area index (LAI) in boreal summer (JJA mean) from the *[Affected LAI]* case; and (b) differences in JJA ozone concentration between the *[Affected LAI]* and *[Intact LAI]* case (i.e., *[Affected LAI]* – *[Intact LAI]*).**

We further investigate the possible pathways contributing to the above simulated ozone changes. Figure 4a shows the JJA ozone dry deposition velocity ($v_d$, cm s$^{-1}$) in the *[Affected LAI]* case, which mirrors the global LAI distribution, reflecting leaf stomatal uptake of ozone. Its absolute changes compared with the *[Intact LAI]* case are shown in Fig. 4b (corresponding percentage changes are shown in Fig. S6), which indicates that regions with a large LAI reduction also have a large decline in ozone dry deposition velocity. The spatial pattern of such a decline is broadly consistent with that of the ozone reduction, suggesting that reduced dry deposition velocity due to ozone-induced LAI decline is an important factor for the higher ozone shown in Fig. 3b.

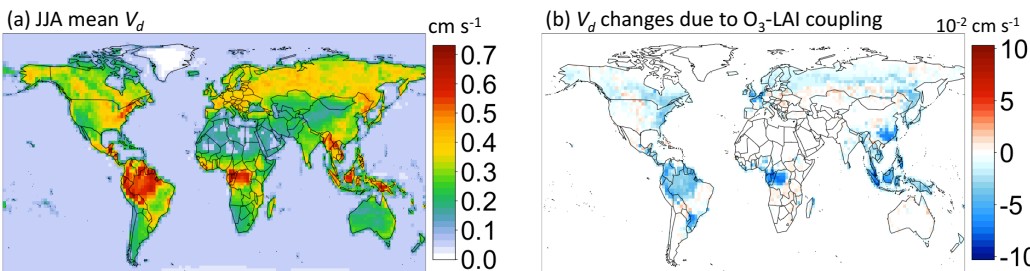

**Figure 4: (a) Ozone dry deposition velocity ($v_d$) in summer (JJA mean) from the *[Affected LAI]* case; and (b) differences in ozone dry deposition velocity between the *[Affected LAI]* and *[Intact LAI]* case (i.e., *[Affected LAI]* – *[Intact LAI]*).**

Global isoprene emission rate (nmol m$^{-2}$ s$^{-1}$) in summer in the *[Affected LAI]* case and its differences from the *[Intact LAI]* case are shown in Fig. 5a and 5b, respectively (percentage changes are shown in Fig. S7). Isoprene, which is one of the most important biogenic VOCs determining ambient ozone, shows a general decline globally, mostly reflecting the strong association between isoprene emission and LAI. Isoprene plays opposite roles in ozone changes depending on the ambient $NO_x$ level. Figure 6 shows the summertime surface $NO_x$ concentration as well as its changes between the *[Affected LAI]* and *[Intact LAI]* case. In the relatively high-$NO_x$ regions at northern midlatitudes (over North America, Europe and East Asia), ozone enhancement from ozone-LAI coupling is relatively small compared with the subtropical and tropical regions (Fig. 3b). This is mostly due to the compensation between the effects of reduced dry deposition, which increases ozone, and reduced isoprene emission, which decreases ozone. In Europe and northern China, in particular, ozone-LAI coupling enhances $NO_x$ level due to reduced sequestration by biogenic VOCs (Fig. 6b), which further limits ozone production due to more sequestration of OH by $NO_x$ (shown in Fig. S8) and thus less efficient cycling of $HO_x$ radicals. On the other hand, in the subtropical and tropical regions where isoprene emission is high (Fig. 5a) and $NO_x$ level is relatively low (Fig. 6a), reduced dry deposition

(Fig. 4b) and reduced isoprene emission (Fig. 5b) add together to enhance ozone concentrations. In central North America, which is low-NO$_x$ in general, the small reduction in NO$_x$ levels is consistent with the slight ozone reduction there (Fig. 3b).

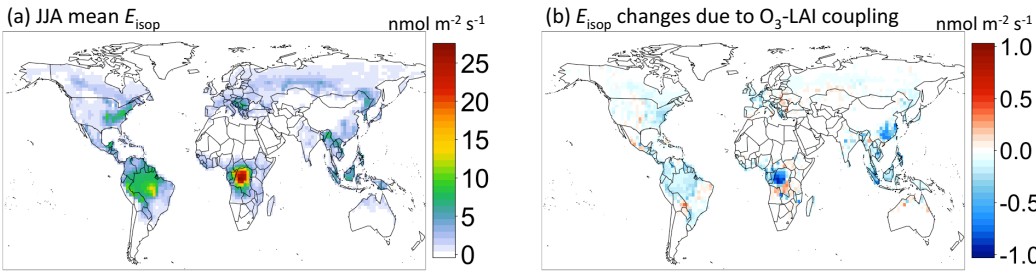

Figure 5: (a) Isoprene emission rate ($E_{isop}$) in summer (JJA mean) from the *[Affected LAI]* case; and (b) differences in isoprene emission rate between the *[Affected LAI]* and *[Intact LAI]* case (i.e., *[Affected LAI]* – *[Intact LAI]*).

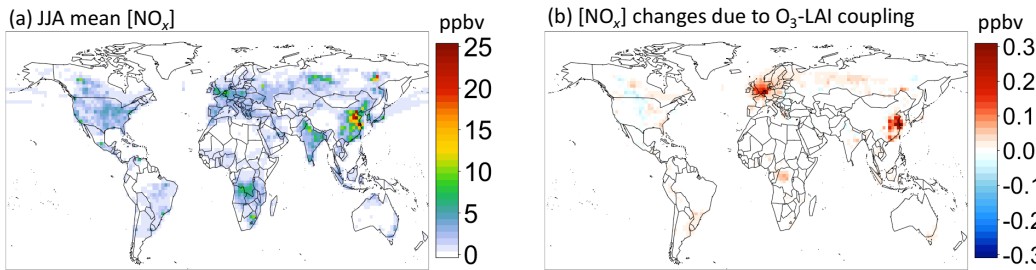

Figure 6: (a) Surface NO$_x$ concentration in summer (JJA mean) from the *[Affected LAI]* case; and (b) differences in NO$_x$ concentration between the *[Affected LAI]* and *[Intact LAI]* case (i.e., *[Affected LAI]* – *[Intact LAI]*).

We further estimate the relative contribution of reduced dry deposition vs. isoprene emission toward the simulated ozone changes under ozone-LAI coupling using the statistical model developed by *Wong et al.* [2018], which is a computationally simple, "offline" assessment tool to estimate the local sensitivity of ozone to any LAI changes, whatever the cause of such changes is, and quantify the relative importance of each of the two dominant pathways (dry deposition vs. isoprene emission) in contributing to this sensitivity as a function of an array of variables including mean ozone concentration, total NO$_x$ emission, wind speed, temperature, etc., for any vegetated locations. According to the statistical model, we find that ozone is substantially enhanced globally when driven only by reduced ozone dry deposition (Fig. 7a), and mildly reduced when driven only by isoprene emission (Fig. 7b). The possible total ozone changes via both pathways combined (Fig. 7c) broadly match the pattern of ozone changes on the continents directly simulated by GEOS-Chem (Fig. 3b). The statistical model suggests that reduced ozone dry deposition plays a more dominant role in the total ozone changes, and reduced biogenic isoprene emission generally causes a decline in ozone that partially offsets the effect of reduced dry deposition especially in

northern midlatitude high-$NO_x$ regions where ozone sensitivity to isoprene emission is stronger than in tropical low-$NO_x$ regions. In general, anthropogenic $NO_x$ emission and baseline LAI are the most important factors determining which of dry deposition vs. biogenic emissions is the dominant pathway accounting for local ozone responses to LAI changes.

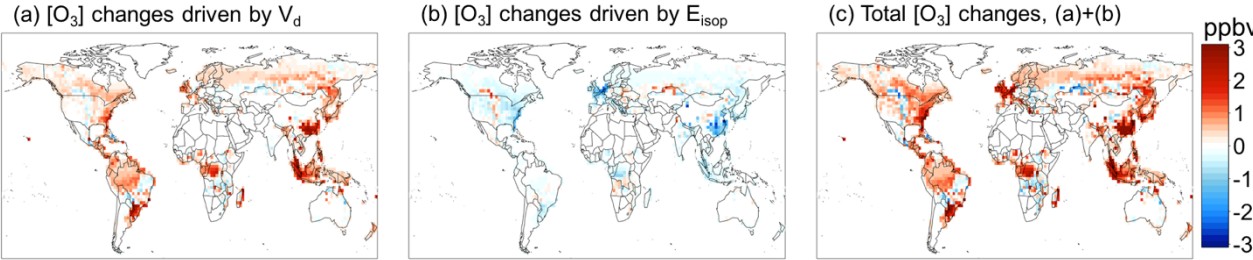

**Figure 7: Attribution of simulated ozone changes to (a) changes in dry deposition only; (b) changes in biogenic isoprene emission only; and (c) changes in both dry deposition and isoprene emission combined, based on the statistical model developed by *Wong et al.* [2018].**

Equivalent plots of Fig. 3 but for boreal wintertime (DJF) mean and annual mean ozone are shown in Fig. S5. Annual ozone changes due to ozone-LAI coupling are broadly consistent with that for JJA mean, albeit with weaker negative feedbacks and more spatially dispersed positive feedbacks. The wintertime ozone enhancements in the Northern Hemisphere are generally much stronger and more widespread than those in summer, mostly due to the smaller importance of isoprene emission
in counteracting the deposition-mediated positive feedbacks.

**5 Impacts of asynchronous ozone-vegetation coupling on surface ozone**

In the above we have presented the effect of synchronous ozone-LAI coupling, whereby ozone and LAI interact dynamically "online" on a monthly timescale according to a simplified parameterization scheme, on ozone air quality. Here
we perform an additional series of GEOS-Chem and CLM experiments to determine the effect of ozone-LAI coupling if the coupling is done asynchronously. The results from these experiments, in comparison with those in Sect. 4, allow us to: 1) examine the relative importance of first-order and second-order feedback effects; 2) check if driving CLM with temporally varying ozone fields would yield different results; and 3) evaluate if the "quasi-steady state response" assumption behind the ozone-LAI synchronous coupling is reasonable.
We first simulate hourly ozone concentrations using GEOS-Chem under year-2012 conditions, which are then used to drive CLM with the ozone damage scheme (Sect. 2.2) for at least 20 simulation years until a quasi-steady state is reached. The resulting relative changes in monthly LAI at PFT levels due to ozone damage, which we can call the "first-order" effect,

are then multiplied by the intact potential LAI and fed into GEOS-Chem to simulate ozone concentrations again, finishing one cycle of coupling. The "new" ozone concentrations are then fed back to CLM to estimate the "new" steady-state LAI changes, which we can call the "second-order" effect. In theory, the feedback cycles should carry on until relative LAI changes and ozone concentrations come into equilibrium with each other. In practice, we find that the second-order LAI changes after one cycle of asynchronous coupling are much smaller than the first-order changes and yield only negligible further changes in ozone concentrations, suggesting that the first-order effect has already encapsulated most of the coupling effect. The final simulated results for both LAI and ozone concentrations should represent a long-term quasi-steady state of dynamic ozone-vegetation interactions.

The asynchronous ozone-LAI coupling experiments have the same model configurations as described in Sect. 2.4 and 2.5, except that $\gamma_{raw}$ in Eq. (7) is now calculated with CLM-simulated LAI driven by hourly ozone fields from the *[Intact LAI]* case, instead of constant ozone levels. We constrain $\gamma_{raw}$ to be within the range of 0.3 to 1.6, which covers more than 90% of its values. In the *[Affected LAI]* case here, we replace $\gamma_{opt}$ in Eq. (10) with $\gamma_{raw}$ to obtain $LAI_{O3}$, which are then used as boundary conditions for GEOS-Chem simulation.

The differences in LAI between the *[Affected LAI]* and *[Intact LAI]* case are shown in Figs. S9 and S10. The mostly negatively relative changes in LAI here for asynchronous coupling are broadly consistent with that shown in Fig. S3 for synchronous coupling, albeit with more frequent occurrences of the sporadic LAI increases in low-LAI regions. This is expected because in the development of parameterization for synchronous ozone-LAI coupling, many of these grid cells are filtered out due to poor fitting by Eq. (8). The overall strong resemblance in the relative LAI changes between the two coupling approaches, at least for regions with sufficiently high LAI, suggests that the simplified parameterization for ozone-LAI coupling on a monthly timescale used in synchronous coupling (Sect. 2.4) is a reasonable idealization of the cumulative long-term steady-state responses of vegetation to temporally varying ozone levels.

Figure 8 shows the JJA mean surface ozone in the *[Affected LAI]* case and the changes from the *[Intact LAI]* case due to asynchronous ozone-LAI coupling. Simulated ozone concentrations are in the same range as that for synchronous coupling with a global average of 28 ppb and highest value of 79 ppb in central Africa. The absolute (Fig. 8b) and relative (Fig. S11) differences in ozone concentrations between the *[Affected LAI]* and *[Intact LAI]* case are broadly consistently with that for synchronous ozone-LAI coupling (Fig. 3b and Fig. S5), but are generally more localized. Similar to the synchronously coupled case, simulated ozone concentrations are higher especially in the tropics, eastern North America, Europe, and southern China by up to 3 ppb, indicating a strong positive feedback due to ozone-LAI interactions. The same figures as Figs. 4-6 but for asynchronous coupling are included in the supplement (Figs. S12-S16), also showing show broadly consistent patterns. Most of the bigger differences occur in low-LAI regions and over the oceans. The discrepancies likely arise from the tendency toward more unstable model (CLM) behaviors at low LAI and the inclusion of diurnally and daily fluctuating ozone in ozone-LAI coupling in the asynchronous approach (as opposed to using constant ozone levels in parameterizing the ozone-LAI relationship). In the original ozone-vegetation scheme in CLM, cumulative ozone damage only occurs when both LAI and ozone level are above some thresholds. Thus, when LAI is low and ozone fluctuating, ozone-LAI coupling becomes more

erratic and loses persistence. Such peculiarities are smoothed out when parameterizing the ozone-LAI relationship by the best-fitting of responses curves and filtering of poorly fitting locations.

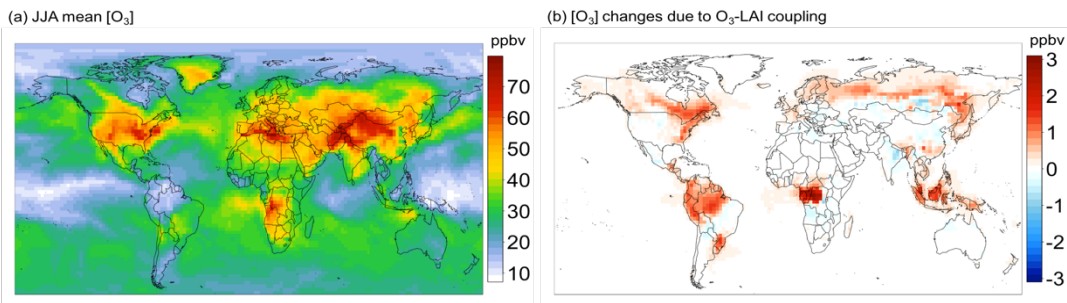

Figure 8: (a) Surface ozone concentration with ozone-affected leaf area index (LAI) in summer (JJA mean) from the *[Affected LAI]* case for asynchronous ozone-LAI coupling; and (b) differences in JJA ozone concentration between the *[Affected LAI]* and *[Intact LAI]* case (i.e., *[Affected LAI]* – *[Intact LAI]*).

## 6 Ozone feedback factor

Climate feedback factor has been widely used in climate studies to indicate how the initial, "direct" surface temperature change driven by a given radiative forcing can be dampened or amplified by internal feedback mechanisms in the climate system [e.g., *Stephens*, 2005]. Here we analogously develop the concept of ozone feedback factor, $f$, which can be used to indicate how an initial ozone change ($\Delta[O_3]_i$) driven by anthropogenic precursor emissions (mostly $NO_x$ and VOCs) can be amplified or dampened by various processes within the earth system (e.g., ozone-vegetation coupling) to arrive at a final ozone change ($\Delta[O_3]_f$):

$$\Delta[O_3]_f = \frac{\Delta[O_3]_i}{1-f} \tag{11}$$

Here, we conduct a simulation, *[Intact_NoAnth]*, in GEOS-Chem with the same settings as the *[Intact LAI]* case (for synchronous coupling) but with all anthropogenic emissions turned off (Table 3). Therefore, the differences between this case and *[Intact LAI]* necessarily represent the effect of anthropogenic forcing (mostly by fossil fuel combustion) on ozone changes without any ozone-LAI feedback, whereas the difference between this case and *[Affected LAI]* represents the "final" effect with ozone-LAI coupling and feedback. In Eq. (11), $\Delta[O_3]_i$ and $\Delta[O_3]_f$ are the differences in ozone concentrations between cases, *[Intact LAI]* – *[Intact_NoAnth]* and *[Affected LAI]* – *[Intact_NoAnth]*, respectively. Thus, analogous to the interpretation of the climate feedback factor, the ozone feedback factor $f$ here reflects the strength and sign of feedback effect from ozone-vegetation coupling on ozone concentration itself, and a value of $f < 0$ represents a negative feedback whereby ozone changes are dampened by ozone-vegetation interactions, and $0 < f < 1$ represents a positive feedback whereby ozone changes are amplified by ozone-vegetation interactions.

The summertime $f$ factor for ozone-LAI coupling based on GEOS-Chem simulations is shown in Fig. 9, where the red areas indicate a positive feedback on ozone concentration after incorporating ozone-LAI coupling and blue areas indicate a negative feedback. We find a significant positive feedback signal in central Africa and the Amazon, which experience a relatively large reduction in ozone dry deposition and isoprene emission due to ozone-LAI coupling (Fig. 4b and Fig. 5b). The strong positive feedback over these tropical forest regions is mostly caused by the combined effects of reduced dry deposition and (to a lesser extent) reduced isoprene emission in a low-$NO_x$ environment, which act in the same direction to increase surface ozone. The negative feedback in many remote regions with low or no LAI, e.g., central North America and most of the Southern Hemisphere, is mostly a result of reduced transport of $NO_x$ reservoir species formed from reactions of $NO_x$ and biogenic VOCs. In contrast, in high-population, high-$NO_x$ regions including the eastern US, Europe and eastern China, the relatively weak positive feedback and even negative feedback (in North China) mostly reflect the compensating effects of reduced isoprene emission (which reduces ozone in high-$NO_x$ regime) and reduced dry deposition (which enhances ozone).

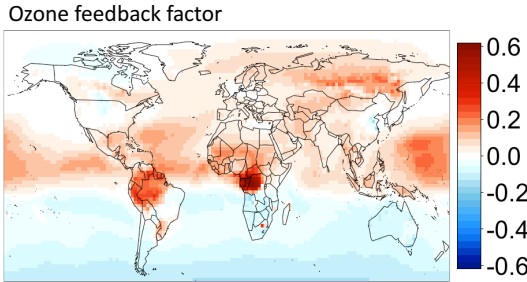

**Figure 9: Ozone feedback factor ($f$) arising from the coupling between surface ozone (JJA mean) and leaf area index (LAI). A value of $0 < f < 1$ indicates a positive feedback (self-amplification of surface ozone), and a value of $f < 0$ indicates a negative feedback (self-dampening of surface ozone).**

Generally, ozone feedback factor of $f > 0$ (positive feedback) occurs in most heavily vegetated areas with low $NO_x$ levels. Distinct ozone positive feedbacks in the tropics indicate a substantial effect from ozone-LAI coupling, which can have important ramifications for future ozone projections. In major populated regions at northern midlatitudes, the relatively weak feedback effect is not a result of the insignificance of every individual pathway, but rather reflects compensating effect between several important pathways.

## 7 Conclusions and discussion

In this study, we develop a parameterized function for an ozone impact factor on LAI by conducting various land surface-biogeochemical model simulations using CLM with an empirical scheme of ozone damage on vegetation to represent the impact of long-term ozone exposure on monthly mean LAI. We then conduct various sets of atmospheric chemical transport

simulations using GEOS-Chem driven by present-day meteorological conditions and PFT fractional coverage under various configurations: with and without parameterized ozone-LAI coupling within GEOS-Chem (ozone-affected LAI vs. intact, potential LAI), with and without anthropogenic emissions, and with synchronous vs. asynchronous ozone-LAI coupling. Such configurations allow us to investigate the impacts of ozone-LAI coupling on simulated ozone air quality and vegetation health, as well as the sign and strength of vegetation-mediated ozone feedback, which can either dampen or amplify the effect of anthropogenic emissions on tropospheric ozone levels.

Generally, ozone damage causes a global LAI reduction for most PFTs under long-term ozone exposure over multiple decades. Compared with the hypothetical intact LAI that is unaffected by ozone, the reduction in PFT-weighted LAI can be as high as 2.6 (percentage reduction up to 50%) in high-LAI regions. Only a few studies of ozone-vegetation interactions have considered this important vegetation structural parameter in coupled model simulations [e.g., *Yue and Unger*, 2014; *Sadiq et al.*, 2017]. The magnitude of simulated LAI changes in this study is quite different from that of *Sadiq et al.* [2017], who used the same ozone damage scheme from *Lombardozzi et al.* [2015] but in a fully coupled land-atmosphere model (CLM4CN-CAM4-Chem) where meteorological variables are also modified dynamically by both stomatal and LAI changes. They found a relatively irregular pattern of summertime LAI changes and the magnitude is generally small (within 5%), likely due to more favorable meteorological conditions triggered by stomatal changes that partly counteract ozone-induced photosynthetic damage (e.g., more convective precipitation and enhanced soil moisture in certain places where surface temperature increases).

Incorporating the ozone-induced damage on LAI in the GEOS-Chem chemical transport model, we find an ozone feedback of –1.8 ppb to +3 ppb globally, and a corresponding ozone feedback factor of about –0.1 to +0.6. The strongest positive feedback from ozone-LAI coupling is found in tropical forests, where dry deposition plays the dominant role modulating the feedback. *Sadiq et al.* [2017] called this kind of feedback "biogeochemical" because it is effected directly through plant ecophysiological responses and allocation to structural parts. *Sadiq et al.* [2017] also considered "biogeophysical" or "meteorological" feedback, whereby ozone-induced damage on plants causes a cascade of meteorological changes that ultimately affect ozone itself. In their study, the total ozone feedback is up to +4-6 ppb, and based on sensitivity simulations they attributed roughly half of that to biogeochemical feedback, which is consistent in both magnitude and sign with this study despite the use of different chemical transport models, although in their study stomatal changes (not considered in this study) play a larger role in the biogeochemical feedback than LAI changes. The remaining positive feedback in their study mostly arises from biogeophysical feedback, whereby reduced stomatal conductance following ozone damage leads to less transpiration, higher vegetation temperature and thus higher isoprene emission.

As our ozone-LAI coupling approach is embedded within a chemical transport model driven by prescribed meteorology, the feedback effects on surface ozone in this study is purely biogeochemical, and decidedly do not include the complication arising from meteorological changes. The feedbacks are attributable to different pathways in different regions. In tropical regions such as maritime Southeast Asia, Amazonia and central Africa, the strong summertime positive ozone feedback is mainly due to the ozone-induced LAI reduction and the subsequent decrease in ozone dry deposition. Reduced isoprene emission further enhances the feedback in these low-$NO_x$ environments but is relatively minor due to the relatively

lower sensitivity of ozone to isoprene emission in these regions. In contrast, in high-$NO_x$ regions such as the eastern US, Europe and eastern China, reduced isoprene emission decreases ozone, and this counteracts with the positive feedback from dry deposition, yielding relatively small or even negative feedback effects there. Over the oceans in the Northern Hemisphere, surface ozone concentration also increases in response to reduced LAI on the continents, mainly because of continental outflow. Over most of the Southern Hemisphere, there is a weak negative feedback, which is likely driven by reduced intercontinental transport of organic nitrate (as a reservoir of $NO_x$) formed from $NO_x$-VOC reactions.

Uncertainty can arise from the large variability in the ozone sensitivity of different plants, especially for tropical trees and grasses, which are modeled based on relatively insufficient data as compared with temperature ecosystems [*Lombardozzi et al.*, 2013]. The current *Lombardozzi et al.* [2015] scheme classifying 15 PFTs into three plant groups is relatively robust in capturing the average plant physiological responses to ozone uptake on a global scale from across many studies, but it treats tropical and temperate plants equivalently as far as ozone sensitivity is concerned, which may lead to possible biases due to an inadequate representation of spatial heterogeneity of plant-ozone ecophysiology. More detailed experimental and field data, especially for tropical and subtropical plants, can potentially help us derive a more region-specific and spatially resolved parameterization that can be particularly useful for high-resolution regional air quality simulations. Along the same line, we find the greatest feedbacks over tropical forests, where ozone concentrations and fluxes are not as well constrained by available in-situ observations as in the midlatitudes. More extensive and long-term measurements of biosphere-atmosphere fluxes in tropical regions are necessary to ascertain the strength of ozone-vegetation feedbacks in these identified hotspots. We also note that our parameterization necessarily ignores hysteresis effect, whereby damage done at incidentally high ozone concentrations may not undergo full recovery even when ozone levels drop again.

This study focuses exclusively on ozone-LAI coupling, but the interaction between ozone and stomatal conductance has also been shown to substantially modulate ozone-vegetation feedbacks [*Sadiq et al.*, 2017]. Ozone-stomata coupling using the same modeling framework certainly warrants further investigation. This study also considers ozone effects on biogenic VOC emissions only via the gradual modification of LAI, but previous studies have suggested that chronic ozone exposure may inhibit isoprene biosynthesis by directly interfering with enzymatic activities [e.g., *Calfapietra et al.*, 2007], and high ozone episodes may even enhance isoprene emission by triggering plant defense mechanisms against oxidative stress [e.g., *Fares et al.*, 2006]. It is necessary to further examine the interactions between ozone and isoprene biosynthesis on different timescales. Moreover, in this study meteorological conditions are prescribed and looped over for a typical year only, and thus the potential impacts of interannual climate variability on the ozone-LAI relationships are not fully considered. For instance, the occurrence of droughts may either weaken or strengthen the coupling between plants and ozone by interfering with photosynthetic capacity, stomatal behaviors and biogenic emissions [e.g., *Wang et al.*, 2017]. Despite the limitations stated above, our findings still attest to the existence of strong ozone biogeochemical feedbacks under typical conditions, and highlight the importance of incorporating ozone-vegetation coupling into regional to global air quality and ecosystem health assessment so that more realistic future projections can be made.

## Data availability

Most of the data produced by this study and presented in the manuscript are deposited in the publicly available institutional repository, accessible via this link: http://www.cuhk.edu.hk/sci/essc/tgabi/data.html. Request for raw data or the complete set of data, or any questions regarding the data, can be directed to the principal investigator, Amos P. K. Tai
(amostai@cuhk.edu.hk).

## Acknowledgement

This work was supported by the Early Career Scheme (Project #: 24300614) of the Research Grants Council of Hong Kong given to the principal investigator, Amos P. K. Tai. We also thank the Information Technology Services Centre (ITSC)
at The Chinese University of Hong Kong for their devotion in providing the necessary computational services for this work. Colette L. Heald acknowledges the support from the National Science Foundation (ATM-1564495) for the development of land cover harmonization used in this work.

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
