# Peer review of "Coupling between surface ozone and leaf area index in a chemical transport model: Strength of feedback and implications for ozone air quality and vegetation health"

_Atmospheric Chemistry and Physics, 2018_

## Referee Comment (RC1) · Anonymous Referee #1 · 18 May 2018

Zhou et al. (2018) develop a parameterization for ambient ozone to impact the LAI that is in turn used in the model parameterization for ozone sources and sinks. The main points of this paper is that the sign and strength of the ozone-LAI feedback depends on regional NOx. Although this paper is timely with respect to the substantial interest in biosphere-atmosphere interactions through atmospheric chemistry, I think this paper needs major revisions. If the authors address these revisions then the paper should be substantive and appropriate for publication in ACP.

What is new about the paper is that the authors isolate the feedback between LAI and

ozone. However, I think the authors' choice to isolate this feedback from the rest of the ozone-vegetation interactions needs to be better justified. The authors say that the other ozone-vegetation interactions are uncertain, but the feedback between LAI and ozone is also uncertain.

The numbers that the authors give for LAI decreases due to ozone are substantial. The ozone damage to stomatal conductance and photosynthesis parameterization developed by Lombardozzi et al. (2012, 2015) is directly constrained from observations, but if the atmospheric chemistry model simulates too much surface ozone then there will be too much ozone damage. How much of the feedback is because there is too much ozone? Further, BVOC emission and ozone dry deposition parameterizations are highly sensitive to changes in LAI. Are the authors confident that these processes actually respond this strongly to LAI?

The authors often discuss regional hotspots of changes in ozone sources and sinks. But perhaps these regions are most poorly constrained in terms of their natural emissions and dry deposition, nonetheless anthropogenic emissions and ambient chemistry. Although this paper could motivate more observations of ozone sources and sinks in these regional hotspots, perhaps the focus of the discussion should be on more regions where ozone can be constrained from observations.

For the asynchronous ozone-vegetation coupling, I understand the authors' use as a sanity check. But what are examples of the first and second order feedbacks that the authors describe? Does this asynchronous coupling allow meteorology to respond to changes in LAI? I'm not sure I understand how their method helps them to address their third goal of "[evaluating] if the "quasi-steady state response" assumption behind the ozone-LAI synchronous coupling is reasonable". This may come from my not understanding the problem as described at the end of Section 2. Will the authors please clarify their statement of the problem?

Further, I think the assumptions going into feedback factor analysis needs to be more

clearly laid out.

Specific comments. I would advise that the authors avoid the term "significant" unless the authors use statistical testing to determine the significance of a change. I would also advise that the authors avoid the term "vegetation-ozone coupling" because as they mention, they are isolating the LAI-ozone coupling.

Page 1. In general, there are many jargon terms and ambiguous descriptors in the abstract. Line 14: From the first sentence to the second, it's not clear that there is an "interdependence" between ozone impacts on vegetation & vegetation impacts on ozone. Line 21: Will the authors use a term other than "correlates" here? Line 21: "dynamically forcing" seems contradictory to me. I think the authors should find another way to describe this. Lines 24-25: Standing alone in the abstract, "ozone feedback" and especially "ozone feedback factor" are not meaningful to readers. The authors should either define them in the abstract or use plainer language in the abstract. Page 2. Line 2: I would say "important ramifications for more realistic assessment of ozone air quality and ecosystem stretch" is a stretch Line 16: Tai et al. (2013) is not in the references. Line 17: "offline-coupled" is a bit ambiguous - can the authors clearly articulate what this means? Lines 22-23: Is this isoprene chemistry current? Lines 24-25: This is not exactly right. The authors should revise their description of the chemical loss pathways of ozone Line 25: Dry deposition does not mainly occur through leaf stomatal uptake. In addition, Wang et al. (1998) is not an appropriate citation here Page 3. Lines 4-5: How are there dynamic changes in PFT distribution following ozone damage? Line 5: Li et al. ERL 2016 does as well Line 15: Lombardozzi et al. (2015) used fixed satellite LAI? It seems strange that LAI would not be coupled into the carbon and water cycles in CLM. Lines 13-27: The point of this paragraph is unclear. Page 4. Line 1-4: How do the authors conclude that these changes in meteorology due to ozone damage are more uncertain, or less important, as the feedbacks between LAI and ozone? Why should we have one and not the other in our CTMs? Are the feedbacks between LAI and ozone realistic if we are not accounting for coupling between vegetation, meteorology

and ozone? Line 10-11: Please specify the prescribed atmospheric data Page 5. Line 12: The authors should elaborate on the "updates on important canopy processes" Line 10: "hydrometeorological variables" is a bit vague Line 18: Sitch et al. (2007) is not really an appropriate reference here Line 25-26: Table 1 gives no indication of how impacts of ozone vary more generally among three groups rather than 15 because the authors only present the three groups Page 6. Line 15-17: Sure, GEOS-Chem has been extensively used and evaluated, but that doesn't mean that it looks good against observations. Some discussion of the surface ozone bias is needed. Lines 21-30: A more detailed description of soil NOx, BVOC and dry deposition parameterized mechanisms would be helpful, especially how LAI fits into them. Note that the authors have previously defined r_a as r_a + r_b. What do "sublayer resistance" and "bulk surface resistance" mean? Silva et al. (2018) find that the ozone dry deposition scheme in GEOS-Chem is highly sensitive to changes in LAI. This means that any changes in LAI are going to impact ozone deposition velocity, but is this strong dependence constrained by observations? Lines 31 (page 6) to 6 (page 7): I'm confused about the harmonization. So PFT distribution is not the same for soil NO emission, BVOC emission, and dry deposition parameterizations? Page 7. Line 9: Are the constant ozone levels prescribed for each grid box? Line 14: What does "one-sided exposed LAI" mean? Why do the authors use it? Lines 23-31: I find the terms "incremental increase" and "incremental decrease" used in this context ambiguous. It might be helpful to give an example in the supplementary text of what the authors are describing on Lines 29-31 (e.g., similar to their Figure 1). Page 8. Line 12: This is a factor of five difference. Why is there such a large range? Line 14: Does this mean that values below 0.3 are set to 0.3 and values above 1.5 are set to 1.5? Page 9. Lines 5-6: It's not clear to me how the authors will use Intact_NoAnth to examine the strength of the ozone feedback. Lines 7-8: Why do the authors need three years after spin up? Shouldn't the spin up be used to reach quasi-steady state? Do the authors average across the three years after spin up, or just use the third 2012? Lines 12-19: This is unclear to me. My impression is that the authors do not want to use observationally-derived LAI here because it would already include the impact of historical ozone damage. So, the authors need to derive a potential LAI. Is this the case? Will the authors spell this out a little more clearly? Why are the authors using CLM MODIS LAI here, rather than the LAI used in GEOS-Chem? I have some qualms with "maximum LAI possible if there is no ozone damage in reality" because of the "self-healing" effect. Lines 20-24: Is LAI only changing on a monthly basis? If so, is there a large change in LAI from May 31 to June 1 over northern midlatitude forests? How does this influence ozone? Could this impact the authors' results? Page 10. Line 3: Typo on this line Line 30: Is this physically constrained by observations? A decrease of up to 2.6 m2/m2 seems really high. On a similar note, the authors' Figure S3 shows relative changes in LAI due to ozone are pretty high. Could the ozone impact on LAI be overstated? Page 12. Lines 3-5: Can the authors show this in a figure in supplemental? Page 13. Lines 5-9: This sentence is a bit unclear. Page 14. It would be helpful to have the same color scale on Figure 7 and Figure 3b. The Wong et al. (2018) statistical technique needs to be described in further detail in the manuscript. Does this technique only use the changes in LAI due to ozone damage archived from GEOS-Chem, or does it also use archived ozone concentrations, isoprene and deposition? Page 15. Line 12: What are the first and second order feedback effects? Page 17. Line 23: The authors should cite the specific chapter in the IPCC (2013) report that they are referring to, not the entire report. Line 24: I don't think it is appropriate to describe the formation of ozone from NOx and VOCs "anthropogenic forcing of precursor emissions" Line 20-26: Does this approach assume that the change in ozone due to the LAI feedback + the change in ozone due to NOx emissions = change in ozone due to both NOx emissions + LAI feedback? I think this needs to be stated and the limitations of the approach discussed. Page 18. Lines 9-10: I'm not sure I understand the difference between "with and without the parameterized ozone-LAI relationship" and "with synchronous vs. asynchronous ozone-LAI coupling". Is the difference that for the first, only GEOS Chem is used, and for the second GEOS Chem and CLM are used? In general, finding a clearer way of describing the different set-ups would be helpful. Line 14: What are the authors defining as long-term here?

Line 17: What is the important vegetation structural parameter? Lines 17-23: I'm a bit confused because the authors say here that the LAI changes are quite different from Sadiq et al. (2017), but then say later on that the "biogeochemical" feedback is similar between the the two studies. Page 19. Lines 2-5: This sentence is a bit unclear. To the best of my understanding, the authors are implying that ozone damage influence on sensible vs. latent heat flux partitioning and the resulting model meteorology is wacky and less certain than the influence of ozone damage on LAI. How do the authors justify this? Lines 24-32: I find this particular discussion confusing. Because LAI evolves on slower timescales, I'm not seeing why it is a problem that LAI is updated to reflect ozone damage on the monthly timescale. To me, the bigger issue is whether the authors are resolving seasonal transitions in LAI on monthly or daily timescales. How do the authors know that the parameterization of ozone-LAI relationship is based on the decadal timescale?

---

## Referee Comment (RC2) · Anonymous Referee #2 · 5 Jul 2018

The main goal of this study is to quantify the impacts of ozone vegetation damage on the atmospheric surface ozone concentrations themselves (ozone air quality). The research team design an intelligent set of systematic global modeling experiments to parse out this particular air quality feedback focused on changes in the LAI only (ignoring meteorology changes). For example, where ozone vegetation damage has been incorporated in coupled chemistry-climate models and Earth system models, (such as NCAR CESM and NASA GISS ModelE2) it is extremely challenging (and maybe impossible) to assess the actual sign and magnitude of this feedback on surface ozone air

quality due to the complex bi-directional linkages between the vegetation, meteorology and atmospheric chemistry. The study builds on a recent previous paper: Sadiq, M., Tai, A. P. K., Lombardozzi, D., and Val Martin, M.: Effects of ozone–vegetation coupling on surface ozone air quality via biogeochemical and meteorological feedbacks, Atmos. Chem. Phys., 17, 3055-3066, https://doi.org/10.5194/acp-17-3055-2017, 2017. The previous paper used sensitivity experiments within a single model framework (CLM-CESM) to examine the underlying driving mechanisms for the ozone-vegetation feedbacks. A previous conclusion was that reduced transpiration leading to increased leaf temperatures (and increased BVOC emissions) is an important mechanism in the NH mid-latitudes leading to a fairly strong positive surface ozone response. The previous work is appropriately discussed in the current paper, and a rationale is provided to focus on the relatively smaller feedbacks through LAI changes only in this work.

The workload represented in this paper is extensive and impressive, including developing an "ozone damage" LAI functional algorithm that is implemented into GEOS-Chem, and multiple synchronous and asynchronous coupling experiments using CLM and GEOS-Chem. The high quality and clarity of the writing and presentation means that it is possible to follow fairly easily the complex experimental design and methodology. The model results are applied to understand the underlying LAI-related biogeochemical mechanisms (dry deposition versus BVOC emissions only here) that drive the ozone-LAI feedback in the current model framework. The major important new findings of the study are that the O3-LAI feedback can have a different sign on surface ozone AQ depending on region and level of NOx pollution; and that the positive feedback is particularly strong in tropical regions. The study also introduces and calculates a new metric "ozone feedback factor" that is strongly positive in tropical regions, which is an additional important contribution to the literature.

1. A major finding and possibly the most interesting aspect of the study is the high sensitivity positive feedback in the tropics (through the reduced dry deposition). I believe that this result is based on the application of ozone damage parameters (photosynthesis and stomatal conductance) for temperate plants (e.g. Lombardozzi et al., 2011; 2013)? Is this correct? The entire model framework assumes that tropical plants behave like temperate zone plants in response to ozone? The paper needs to emphasize more strongly that there are essentially no ozone sensitivity measurement data for tropical plants, and therefore the implications for the value of the results.

2. The exponential LAI parameterization function for GEOS-Chem. Based on Figure 1, the saturation occurs for relatively low ozone. Between 40 and 100 ppbv there is no dependence of LAI on ozone concentration. This function seems to be physically unrealistic. We would expect the LAI response of a forest or cropland ecosystem growing in ~45 ppbv ambient ozone to be rather different to such in ~90 ppbv ambient ozone?

3. Regarding the LAI function. The paper could be greatly strengthened by showing validation and evaluation of the LAI function against measurement data (or even plant biomass could be used as proxy for some ecosystems where LAI changes are less available).

4. The paper assumes that BVOC emissions are essentially positive linear function of LAI. In reality, ozone vegetation damage may influence BVOC emissions in complex ways (even independent of LAI) through changes in biochemistry and plant production, and even lead to increases in BVOC emissions. There is a growing literature in this area that needs to be cited and discussed. The paper needs to emphasize the limitations of the BVOC modeling response and that the BVOC response sign could be different on monthly timescales (positive versus negative).

5. The study seems to only consider isoprene? I agree isoprene is by far the most important for ozone, but how do changes in other BVOCs influence the ozone-LAI feedbacks? For example, monoterpenes and sesquiterpenes in the tropics? Both CLM and GEOS-Chem do include higher level BVOCs and terpenes.

6. There are some curious features of Figure 3a, the baseline surface ozone distribution, in this GEOS-Chem model. For example, (i) surface ozone in eastern China

is about 40-50 ppbv, much lower (about half of the levels) than in Northeastern US (70-80 ppbv). Indian subcontinent has very low ozone whereas Sahara Desert has substantially higher ozone. The highest European values in summer are over the Mediterranean Sea rather than the continental land mass. Do these features agree with current ozone measurement monitoring networks in these regions?

7. Related to Figure 3b, the authors offer an explanation for the decreased ozone signals in US crop belt and North China Plain: "Such a reduction is driven by reduced transport of VOCs as well as organic nitrate formed from VOC-NOx reactions following reduced LAI elsewhere in more vegetated regions". For sure, their model shows limited to no LAI changes in these regions. However, it is really interesting that these regions are heavily dominated by crop ecosystems where we would expect to see substantial relative changes in LAI due to ozone damage in reality. Are specific crop types represented in the CLM model version? Would the results be different in sign if specific crop types are represented in the model?

8. The difference between the synchronous and asynchronous coupling methods (Figure 3b versus 8b) are massive for the [O3] changes due to O3-LAI coupling. At least the sign is the same, but the spatial responses are very different, especially over N America, Europe, Central Asia, Middle East, N Africa and E China. In many of these widespread regions, synchronous shows a strong signal, but asynchronous has no signal. The authors state: "Most of the bigger differences occur in low-LAI regions which are more prone to idiosyncratic model (CLM) behaviors and numerical outliers especially in the asynchronously coupled cases where such peculiarities are not smoothed out." The paper needs to offer a more scientific, and more physically mechanistic explanation for these differences (rather than "idiosyncratic model (CLM) behaviors"). What does "smoothed out" mean? How can readers know which is the most realistic response? Please directly link the results to the 3 reasons for doing the asynchronous experiments (Page 15, Lines 10-15).

Minor comments Fig. 7 Title "Attribution"

---

## Author Comment (AC1) · 30 Aug 2018

A PDF version of the author responses to referees' comments with proper formatting to clearly indicate referees' comments, our responses, cited revised texts and specialized symbols is included as a supplement. Although the full responses are also copied and pasted below, some formatting is lost, and we highly recommend the Editor to use the PDF version for a more convenient perusal.

####################################

[Figure]

Our point-by-point responses are provided below. The referees' comments are italicized, our new/modified text is highlighted in bold. The revised manuscript with tracked changes is also included in the linked file below for the Editor's easy reference: https://www.dropbox.com/s/ejy5uuo3kafrip6/Zhou_2018_revised_v1.docx?dl=0

Response to Referee #1

Referee: Zhou et al. (2018) develop a parameterization for ambient ozone to impact the LAI that is in turn used in the model parameterization for ozone sources and sinks. The main points of this paper is that the sign and strength of the ozone-LAI feedback depends on regional NOx. Although this paper is timely with respect to the substantial interest in biosphere-atmosphere interactions through atmospheric chemistry, I think this paper needs major revisions. If the authors address these revisions then the paper should be substantive and appropriate for publication in ACP.

We thank the reviewer for the very helpful comments. The paper has been revised substantially to address the reviewer's concerns point by point, and all changes are cited and discussed in the responses below.

Referee: What is new about the paper is that the authors isolate the feedback between LAI and ozone. However, I think the authors' choice to isolate this feedback from the rest of the ozone-vegetation interactions needs to be better justified. The authors say that the other ozone-vegetation interactions are uncertain, but the feedback between LAI and ozone is also uncertain.

The reviewer aptly pointed out that all individual pathways behind ozone-vegetation coupling are themselves uncertain. This paper, therefore, does not argue that ozone-LAI interaction is any less uncertain, but attempts to tease out this particular pathway (i.e., via LAI changes without meteorological feedbacks) in a modeling framework that can assess its relative contribution to the full ozone-vegetation coupling. This contributes to an overall goal to identify the dominant pathways to focus on in future research. Our focus on LAI first is motivated by its ubiquitous importance as a key land

surface parameter in any atmospheric chemistry models (articulated also in P3 L18). We now make these points more clearly:

P3 L17: "... to transpiration changes. This complication from meteorological feedbacks could mask the relative importance of individual vegetation variables (e.g., LAI) in contributing to the overall coupling effect, rendering attribution more difficult."

P3 L19: "LAI is a ubiquitously important land surface parameter driving atmospheric chemistry and hydroclimate in many models [e.g., Wong et al., 2018]. Previous modeling studies ..."

P4 L1: "In this study, we focus on the relationship between ozone and LAI, and assess its relative importance in the overall ozone-vegetation coupling effect. We use a standalone..."

P4 L9: "... This effort not only allows ozone-vegetation coupling to be considered dynamically within an atmospheric model without the complication from meteorological changes and feedbacks, but also..."

We also remove a sentence in the Conclusions and Discussion to avoid overstating the uncertainty of biogeophysical pathways in comparison with the biogeochemical pathways: "... whereby reduced stomatal conductance following ozone damage leads to less transpiration, higher vegetation temperature and thus higher isoprene emission. However, they noted that dynamically changing hydrometeorological variables and processes are not generally tuned for atmospheric chemical measurements, and may introduce large uncertainty in simulating and attributing boundary-layer ozone concentrations to various determining factors."

Referee: The numbers that the authors give for LAI decreases due to ozone are substantial. The ozone damage to stomatal conductance and photosynthesis parameterization developed by Lombardozzi et al. (2012, 2015) is directly constrained from observations, but if the atmospheric chemistry model simulates too much surface ozone

then there will be too much ozone damage. How much of the feedback is because there is too much ozone? Further, BVOC emission and ozone dry deposition parameterizations are highly sensitive to changes in LAI. Are the authors confident that these processes actually respond this strongly to LAI?

The majority of the feedback indeed occurs when ozone level goes from zero (no ozone damage at all) to an ambient level. As can be seen from the LAI-to-ozone response curve (Fig. 1), at ambient ozone levels (e.g., > 30 ppbv) LAI responds only tenuously to any more perturbation in ozone level. Therefore, the general high biases in simulated ozone by many atmospheric chemistry models (e.g., Lamarque et al., 2012) relative to observed ambient ozone levels would contribute only little to the total feedback. The shape of the LAI-to-ozone response curve is now discussed more:

P10 L25: "...  Moreover, the month-to-month variations in ozone concentration are typically much smaller than an incidental jump between zero and a prescribed level. As shown by Fig. 1, the strongest response of LAI to ozone happens at low ozone levels, while at and above ambient levels (e.g., > 30 ppb), LAI responses to any ozone variations also become progressively small. We thus assume that the "steady-state response" for ozone-LAI relationship is reasonably robust to represent short-term responses of LAI superimposed on the long-term, ozone-induced decline in seasonally varying LAI. ... "

The sensitivities of BVOC emissions and dry deposition velocities to LAI changes are assessed using the statistical model of Wong et al. [2018], as discussed in P14 L12 – P15 L3. The statistical model is also based on GEOS-Chem parameterizations and simulations. The LAI dependence in these parameterizations is typically well constrained theoretically by canopy biophysics and empirically by flux observations.

Referee: The authors often discuss regional hotspots of changes in ozone sources and sinks. But perhaps these regions are most poorly constrained in terms of their natural emissions and dry deposition, nonetheless anthropogenic emissions and ambient

chemistry. Although this paper could motivate more observations of ozone sources and sinks in these regional hotspots, perhaps the focus of the discussion should be on more regions where ozone can be constrained from observations.

We agree with the reviewer that the strongest feedback we find occurs where ozone concentrations and fluxes are not well constrained by observations, i.e., in the tropical forests. We now address this in the Conclusions and Discussion:

P20 L20: "... for high-resolution regional air quality simulations. Along the same line, we find the greatest feedbacks over tropical forests, where ozone concentrations and fluxes are not as well constrained by available in-situ observations as in the midlatitudes. More extensive and long-term measurements of biosphere-atmosphere fluxes in tropical regions are necessary to ascertain the strength of ozone-vegetation feedbacks in these identified hotspots. We also note that ..."

In the rest of the paper, we also dissect and discuss extensively the feedback pathways in midlatitude high-NOx regions, especially in North America and Europe, where ozone concentrations and fluxes are better constrained than in the tropics.

Referee: For the asynchronous ozone-vegetation coupling, I understand the authors' use as a sanity check. But what are examples of the first and second order feedbacks that the authors describe? Does this asynchronous coupling allow meteorology to respond to changes in LAI? I'm not sure I understand how their method helps them to address their third goal of "[evaluating] if the "quasi-steady state response" assumption behind the ozone-LAI synchronous coupling is reasonable". This may come from my not understanding the problem as described at the end of Section 2. Will the authors please clarify their statement of the problem?

First and second order feedbacks refer to the number of sequential feedback loops, or "cycles", considered in an asynchronous coupling experiment. They have no additional physical meaning other than in the context of modeling experiments, and in our experiments meteorology is never allowed to respond to LAI changes. We now explain

it more clearly:

P15 L27: "... The resulting relative changes in monthly LAI at PFT levels due to ozone damage, which we can call the "first-order" effect, are then multiplied by the intact potential LAI and fed into GEOS-Chem to simulate ozone concentrations again, finishing one cycle of coupling. The "new" ozone concentrations are then fed back to CLM to estimate the "new" steady-state LAI changes, which we can call the "second-order" effect. In theory, the feedback cycles should carry on until relative LAI changes and ozone concentrations come into equilibrium with each other. ..."

The asynchronous coupling experiment is designed to decidedly represent a long-term quasi-steady state of dynamic ozone-vegetation coupling. If the simulated responses are similar to those from the synchronous coupling experiment, then we may at least assume that the parameterization used in the latter is a reasonable representation of the a quasi-steady-state response. These points are now discussed in greater detail:

P10 L25: "... Moreover, the month-to-month variations in ozone concentration are typically much smaller than an incidental jump between zero and a prescribed level. As shown by Fig. 1, the strongest response of LAI to ozone happens at low ozone levels, while at and above ambient levels (e.g., > 30 ppb), LAI responses to any ozone variations also become progressively small. We thus assume that the "steady-state response" for ozone-LAI relationship is reasonably robust to represent short-term responses of LAI superimposed on the long-term, ozone-induced decline in seasonally varying LAI. This assumption is further tested by driving CLM with hourly varying GEOS-Chem ozone fields in an asynchronous coupling experiment until LAI reaches a quasi-steady state, which will be discussed in Sect. 5."

P16 L6: "... suggesting that the first-order effect has already encapsulated most of the coupling effect. The final simulated results for both LAI and ozone concentrations should represent a long-term quasi-steady state of dynamic ozone-vegetation interactions."

P16 L18: "... The overall strong resemblance in the relative LAI changes between the two coupling approaches, at least for regions with sufficiently high LAI, suggests that the simplified parameterization for ozone-LAI coupling on a monthly timescale used in synchronous coupling (Sect. 2.4) is a reasonable idealization of the cumulative long-term steady-state responses of vegetation to temporally varying ozone levels."

Referee: Further, I think the assumptions going into feedback factor analysis needs to be more clearly laid out.

The feedback factor analysis mainly serves as an alternative and arguably novel way to examine ozone feedbacks, analogous to the analysis of climate feedbacks, based on the simulated results from a set of model experiments. The assumptions behind the analysis do not differ from those behind any model experiments we have conducted. We have, however, attempted to explain the feedback factor in clearer wording: P17 L18: "Here, we conduct a simulation, [Intact_NoAnth], in GEOS-Chem with the same settings as the [Intact LAI] case (for synchronous coupling) but with all anthropogenic emissions turned off (Table 3). Therefore, the differences between this case and [Intact LAI] necessarily represent the effect of anthropogenic forcing (mostly by fossil fuel combustion) on ozone changes without any ozone-LAI feedback, whereas the difference between this case and [Affected LAI] represents the "final" effect with ozone-LAI coupling and feedback. In Eq. (11), ïĄĎ[O3]i and ïĄĎ[O3]f are the differences in ozone concentrations between cases, [Intact LAI] – [Intact_NoAnth] and [Affected LAI] – [Intact_NoAnth], respectively. Thus, analogous to the interpretation of the climate feedback factor, the ozone feedback factor f here reflects the strength and sign of feedback effect from ozone-vegetation coupling on ozone concentration itself, and a value of f < 0 represents a negative feedback whereby ozone changes are dampened by ozone-vegetation interactions, and 0 < f < 1 represents a positive feedback whereby ozone changes are amplified by ozone-vegetation interactions."

Specific comments: Referee: I would advise that the authors avoid the term "significant" unless the authors use statistical testing to determine the significance of a

change. I would also advise that the authors avoid the term "vegetation-ozone coupling" because as they mention, they are isolating the LAI-ozone coupling.

Where appropriate, "ozone-LAI coupling" is used if we are referring specifically to the results from our model experiments, and "ozone-vegetation coupling" is used if we are referring to ozone-vegetation interactions in general, which can include other factors than LAI alone. We have also trimmed down the use of "significant" in response to the reviewer's concern, e.g., replacing it with "substantial" or "important" in many places where it has no statistical inference. It remains where it does have statistical meaning.

Referee: Page 1. In general, there are many jargon terms and ambiguous descriptors in the abstract.

The abstract has now been modified to use less jargons and more discrete terms: "Tropospheric ozone is an air pollutant with substantial harm on vegetation, and is also strongly dependent on various vegetation-mediated processes. The interdependence between ozone and vegetation may constitute feedback mechanisms that can alter ozone concentration itself but have not been considered in most studies to date. In this study we examine the importance of dynamic coupling between surface ozone and leaf area index (LAI) in shaping ozone air quality and vegetation. We first implement an empirical scheme for ozone damage on vegetation in the Community Land Model (CLM), and simulate the steady-state responses of LAI to long-term exposure to a range of prescribed ozone levels (from 0 ppb to 100 ppb). We find that most plant functional types suffer a substantial decline in LAI as ozone level increases. Based on the CLM-simulated results, we develop and implement in the GEOS-Chem chemical transport model a parameterization that computes fractional changes in monthly LAI as a function of local mean ozone levels. By forcing LAI to respond to ozone concentrations on a monthly timescale, the model simulates ozone-LAI coupling dynamically via biogeochemical processes including biogenic volatile organic compound (VOC) emissions and dry deposition, without the complication from meteorological changes. We find that ozone-induced damage on LAI can lead to changes in ozone concentrations

by −1.8 ppb to +3 ppb in boreal summer, with a corresponding ozone feedback factor of −0.1 to +0.6 that represents an overall self-amplifying effect from ozone-LAI coupling. Substantially higher simulated ozone due to strong positive feedbacks is found in most tropical forests, mainly due to the ozone-induced reductions in LAI and dry deposition velocity, whereas reduced isoprene emission plays a lesser role in these low-NOx environments. In high-NOx regions such as eastern US, Europe and China, however, the feedback effect is much weaker and even negative in some regions, reflecting the compensating effects of reduced dry deposition and reduced isoprene emission (which reduces ozone in high-NOx environments). In remote, low-LAI regions including most of the Southern Hemisphere, the ozone feedback is generally slightly negative due to the reduced transport of NOx-VOC reaction products that serve as NOx reservoirs. This study represents the first step to account for dynamic ozone-vegetation coupling in a chemical transport model with ramifications for a more realistic joint assessment of ozone air quality and ecosystem health."

Referee: Line 14: From the first sentence to the second, it's not clear that there is an "interdependence" between ozone impacts on vegetation & vegetation impacts on ozone.

The interdependence stems from the first sentence, "Tropospheric ozone is an air pollutant with substantial harm on vegetation, and is also strongly dependent on various vegetation-mediated processes."

Referee: Line 21: Will the authors use a term other than "correlates" here?

We now use ". . . that computes . . . as a function of . . ." instead. See the revised abstract above.

Referee: Line 21: "dynamically forcing" seems contradictory to me. I think the authors should find another way to describe this.

The wording is now changed. See the revised abstract above.

[Figure]

Referee: Lines 24-25: Standing alone in the abstract, "ozone feedback" and especially "ozone feedback factor" are not meaningful to readers. The authors should either define them in the abstract or use plainer language in the abstract.

The wording is now changed. See the revised abstract above.

Referee: Page 2. Line 2: I would say "important ramifications for more realistic assessment of ozone air quality and ecosystem stretch" is a stretch

The wording is now changed. See the revised abstract above.

Referee: Line 16: Tai et al. (2013) is not in the references.

The reference is now added.

Referee: Line 17: "offline-coupled" is a bit ambiguous - can the authors clearly articulate what this means?

The modeling approach differentiates between synchronous and asynchronous coupling. To avoid confusion, we have now removed "offline".

Referee: Lines 22-23: Is this isoprene chemistry current?

We believe the statement concerned captures the broad understanding of isoprene chemistry to date, without having to specify the nuanced pathways that are still constantly updated.

Referee: Lines 24-25: This is not exactly right. The authors should revise their description of the chemical loss pathways of ozone

The description is modified accordingly:

P2 L23: "... On the other hand, the major sinks for surface ozone include in-situ chemical loss mainly via photolysis and the subsequent reaction of singlet oxygen atom O(1D) with water vapor (H2O), and the dry deposition of ozone onto vegetated surfaces [Wang et al., 1998; Wild, 2007]. ..."

Referee: Line 25: Dry deposition does not mainly occur through leaf stomatal uptake. In addition, Wang et al. (1998) is not an appropriate citation here

For ozone, stomatal uptake is an important component (likely more than half) of the total dry-depositional sink. To be more precise and in response to the reviewer's concern, we have modified that paragraph substantially:

P2 L23: "... On the other hand, the major sinks for surface ozone include in-situ chemical loss mainly via photolysis and the subsequent reaction of singlet oxygen atom O(1D) with water vapor (H2O), and the dry deposition of ozone onto vegetated surfaces [Wang et al., 1998; Wild, 2007]. Leaf stomatal uptake of ozone, in particular, represents 40–60% of the total dry-depositional sink [Fowler et al., 2009]. Vegetation also controls transpiration, which modulates boundary-layer mixing, temperature, water vapor content, and thus the production, dilution and loss of ozone. Therefore, via biogenic VOC emissions, dry deposition and transpiration, vegetation can substantially influence surface ozone concentrations."

The two references, Wang et al. [1998] and Wild [2007], are cited for their estimates of the global tropospheric ozone budget, of which chemical loss and dry deposition are the two major loss pathways. The references are relocated to reflect this better.

Referee: Page 3. Lines 4-5: How are there dynamic changes in PFT distribution following ozone damage?

Ozone damage can influence PFT distribution over time due to the differential abilities of different species to tolerate ozone. We now include this discussion:

P3 L4: "... with the greatest damage (20–25% for GPP, 15–20% for transpiration) happening at northern midlatitudes [Lombardozzi et al., 2015]. The ozone-induced decrease in transpiration has been shown to enhance regional temperature by up to 3°C and reduce precipitation by up to 2 mm d-1 in summertime central US [Li et al., 2016]. Differential abilities of plant species to tolerate ozone, when integrated over space and

time, can also cause the long-term shifts in species richness and ecosystem composition [e.g., Fuhrer et al., 2016]. As vegetation variables such as stomatal resistance, LAI, and plant functional type (PFT) distribution all play important roles shaping surface ozone, dynamic changes in these variables following ozone damage may thus induce a cascade of feedbacks that ultimately affect ozone itself. . . ."

Referee: Line 5: Li et al. ERL 2016 does as well

Li et al. [2016] is now discussed and cited. See our modification right above for P3 L4.

Referee: Line 15: Lombardozzi et al. (2015) used fixed satellite LAI? It seems strange that LAI would not be coupled into the carbon and water cycles in CLM.

Yes, Lombardozzi et al. (2015) used the "SP" (satellite phenology) mode of CLM with fixed, prescribed monthly LAI with daily interpolation. The lack of LAI evolution there is indeed part of the motivation behind our work.

Referee: Lines 13-27: The point of this paragraph is unclear.

The paragraph aims to discuss the importance and challenges of using LAI as a metric for characterizing ozone-vegetation interactions. We now modify the opening sentences to make it clearer:

P3 L19: "LAI is a ubiquitously important land surface parameter driving atmospheric chemistry and hydroclimate in many models [e.g., Wong et al., 2018]. Previous modeling studies of ozone damage on vegetation usually used prescribed LAI and other structural variables such as canopy height derived from satellites and land surveys as fixed parameters in their vegetation simulations [e.g., Sitch et al., 2007; Lombardozzi et al., 2015]. . . ."

Referee: Page 4. Line 1-4: How do the authors conclude that these changes in meteorology due to ozone damage are more uncertain, or less important, as the feedbacks between LAI and ozone? Why should we have one and not the other in our CTMs? Are the feedbacks between LAI and ozone realistic if we are not accounting for coupling

between vegetation, meteorology and ozone?

The gist of this study is to identify the relative importance of individual pathways to guide future research focus. The full response to the reviewer's concern here is already included in our earlier response to the first concern of the reviewer.

Referee: Line 10-11: Please specify the prescribed atmospheric data

Done. P4 L17: "... by prescribed atmospheric data from Climate Research Unit (CRU)–National Centers for Environmental Prediction (NCEP) at a resolution of 1.9ïĆř latitude by 2.5ïĆř longitude. ..."

Referee: Page 5. Line 12: The authors should elaborate on the "updates on important canopy processes"

The updates are specified on P4 L19: "... This version not only updates important canopy processes including canopy radiation and upscaling of leaf processes from previous versions, but also improves the stability of the iterative solution in the computation of photosynthesis and stomatal conductance [Sun et al., 2012]. ..."

Referee: Line 10: "hydrometeorological variables" is a bit vague

We have now removed that phrase altogether. P5 L15: "... Lombardozzi et al. [2012] showed that modifying photosynthesis and stomatal conductance independently using different ozone impact factors can improve model simulations of vegetation responses to ozone exposure. ..."

Referee: Line 18: Sitch et al. (2007) is not really an appropriate reference here

The reference is now removed. Appropriate references are added.

P5 L25: "... CUO is only accumulated when LAI is above a minimum value of 0.5 and ozone flux is larger than a critical threshold of 0.8 nmol O3 m-2 s-1 to account for the compensating ability of plants to detoxify ozone [Lombardozzi et al., 2012; 2015]."

Referee: Line 25-26: Table 1 gives no indication of how impacts of ozone vary more generally among three groups rather than 15 because the authors only present the three groups

We agree with the reviewer that the table gives no indication of the within-group variability of plant responses to ozone. A single set of parameters was obtained for each broad plant group by Lombordozzi et al. [2015], who pulled together data for mostly temperature species from across the world and derived such parameterized relationships. The limitations arising from data paucity are discussed in greater detail in the Conclusions and Discussion:

P20 L13: "Uncertainty can arise from the large variability in the ozone sensitivity of different plants, especially for tropical trees and grasses, which are modeled based on relatively insufficient data as compared with temperature ecosystems [Lombardozzi et al., 2013]. The current Lombardozzi et al. [2015] scheme classifying 15 PFTs into three plant groups is relatively robust in capturing the average plant physiological responses to ozone uptake on a global scale from across many studies, but it treats tropical and temperate plants equivalently as far as ozone sensitivity is concerned, which may lead to possible biases due to an inadequate representation of spatial heterogeneity of plant-ozone ecophysiology. More detailed experimental and field data, especially for tropical and subtropical plants, can potentially help us derive a more region-specific and spatially resolved parameterization that can be particularly useful for high-resolution regional air quality simulations. . . ."

Referee: Page 6. Line 15-17: Sure, GEOS-Chem has been extensively used and evaluated, but that doesn't mean that it looks good against observations. Some discussion of the surface ozone bias is needed.

More discussion of the ozone biases is now included.

P6 L22: ". . . In general, GEOS-Chem underestimates tropospheric ozone in the tropics but overestimates it in the northern subtropics and southern midlatitudes [Zhang et al.,

2010]. For regional surface ozone, the model has small systematic biases overall in the US and China, but has a tendency to overestimate summertime concentrations in the eastern US and certain sites in China [Wang et al., 2009; Wang et al., 2011; Zhang et al., 2011]. . . ."

Referee: Lines 21-30: A more detailed description of soil NOx, BVOC and dry deposition parameterized mech- anisms would be helpful, especially how LAI fits into them. Note that the authors have previously defined r_a as r_a + r_b. What do "sublayer resistance" and "bulk surface resistance" mean? Silva et al. (2018) find that the ozone dry deposition scheme in GEOS-Chem is highly sensitive to changes in LAI. This means that any changes in LAI are going to impact ozone deposition velocity, but is this strong dependence constrained by observations?

CLM calculates resistances at the leaf level and then integrates them to the canopy level via canopy scaling. GEOS-Chem uses parameterization at the canopy level throughout, and therefore the exact definitions of resistances are subtly different between GEOS-Chem and CLM. For instance, the sublayer resistance in GEOS-Chem is equivalent to the canopy-integrated leaf-level boundary layer resistance in CLM. To avoid confusion, we now redefine rǎa and rb in Eq. (3) and the description below:

P5 L23: "where [O3] is the surface ozone concentration (nmol m-3), k_(O_3 ) = 1.67 is the ratio of leaf resistance to ozone to leaf resistance to water, rs here is the leaf-level stomatal resistance, rb is the leaf boundary layer resistance, and ra is the aerodynamic resistance between the leaf and the reference level, . . ."

We also relabel the corresponding canopy-level resistances in GEOS-Chem with capital letters instead of small letters, and include the original description of what bulk surface resistance means. Moreover, the sensitivities of BVOC emissions and dry deposition velocities to LAI changes are assessed using the statistical model of Wong et al. [2018], as discussed in P14 L12 – P15 L3. The LAI dependence in these parameterizations is typically well constrained theoretically by canopy biophysics and empirically

by flux observations. We now also reference this paper again here.

P7 L2: "... whereby dry deposition velocity is the inverse of aerodynamic resistance (Ra), sublayer resistance (Rb) and bulk surface resistance (Rc) added in series. The term Rc accounts for a combination of resistances from vegetation (including stomatal resistance), lower canopy and ground, which have specific values for 11 different land types. Wong et al. [2018] extensively evaluated the LAI dependence of both MEGAN biogenic emissions and dry deposition in GEOS-Chem. ..."

Referee: Lines 31 (page 6) to 6 (page 7): I'm confused about the harmonization. So PFT distribution is not the same for soil NO emission, BVOC emission, and dry deposition parameterizations?

The reviewer is right. Historically, GEOS-Chem has been using different land/plant type classification schemes for different modules, which has been a source of internal inconsistencies. The model is now moving toward harmonizing all land/plant type classifications, but what is important about the Geddes et al. [2016] is that it enables users to input any customized land/plant type distribution and have it harmonized across all modules.

Referee: Page 7. Line 9: Are the constant ozone levels prescribed for each grid box?

Yes.

Referee: Line 14: What does "one-sided exposed LAI" mean? Why do the authors use it?

Across the literature LAI usually means one-sided LAI, but for various purposes CLM distinguishes between one-sided vs. two-sided LAI. Therefore, we specify it here mostly for the reference for CLM users. Exposed LAI accounts for the covering of leaves by snow in winter, but is essentially the same as total LAI in snow-free summer. To avoid confusion (and the paper is mostly concerned with northern summer anyway), "exposed" now only appears once and ELAI is replaced by LAI elsewhere.

Referee: Lines 23-31: I find the terms "incremental increase" and "incremental decrease" used in this context ambiguous. It might be helpful to give an example in the supplementary text of what the authors are describing on

We agree with the reviewer that "incremental" is not the best word to describe what we mean. We now rephrase the relevant sentences to make the meaning clearer.

P8 L4: "... but its decrease per unit ozone increase becomes smaller at higher ozone concentrations (Fig. 1) because of the progressive closure of stomata as represented by the ozone damage scheme. ... In some places especially in grasslands and semi-arid regions, however, $\gamma$raw increases with ozone level but its increase per unit ozone increase also declines and then levels off at higher ozone concentrations."

Referee: Lines 29- 31 (e.g., similar to their Figure 1). Referee: Page 8. Line 12: This is a factor of five difference. Why is there such a large range?

This represents the variability of responses for a different PFTs across a wide variety of environmental conditions and constraints. Despite the large range, a majority of the $\gamma\infty$ values fall within 0.7 and 1. We now specify the variability in greater detail.

P8 L24: "... We find that for 90% of the grid cells/PFTs, $\gamma\infty$ is between about 0.3 and 1.5; for 50% of the grid cells/PFTs, $\gamma\infty$ is between about 0.75 and 0.95. ..."

Referee: Line 14: Does this mean that values below 0.3 are set to 0.3 and values above 1.5 are set to 1.5?

Yes.

Referee: Page 9. Lines 5-6: It's not clear to me how the authors will use Intact_NoAnth to examine the strength of the ozone feedback.

This is explained in detail in Sect. 6 of the revised manuscript.

Referee: Lines 7-8: Why do the authors need three years after spin up? Shouldn't the spin up be used to reach quasi-steady state? Do the authors average across the three

years after spin up, or just use the third 2012?

We decidedly hope to exclude interannual variability from our feedback analysis. Changes from 2011 to 2012 conditions may lead to small perturbations in the ozone-vegetation steady state, and thus after spinning up we let the ozone-vegetation coupling to further settle down to 2012 conditions.

Referee: Lines 12-19: This is unclear to me. My impression is that the authors do not want to use observationally-derived LAI here because it would already include the impact of historical ozone damage. So, the authors need to derive a potential LAI. Is this the case? Will the authors spell this out a little more clearly? Why are the authors using CLM MODIS LAI here, rather than the LAI used in GEOS-Chem? I have some qualms with "maximum LAI possible if there is no ozone damage in reality" because of the "self-healing" effect.

Yes, this is the case, and the reviewer's understanding is correct. The meaning is now spelled out more clearly.

P10 L2: "For the [Intact LAI] case, we first derive an intact, potential LAI that should represent the maximum LAI possible if there is no ozone damage in reality, which is taken as the baseline case for investigating the effect of ozone-LAI coupling. This is necessary because the present-day satellite-derived "observed" LAI is supposedly already the outcome of long-term ozone-vegetation interactions, and should not be used as the baseline. Instead, the potential LAI is used as the initial condition to drive the GEOS-Chem simulations. . . ."

Default MODIS LAI in GEOS-Chem is available at the total, grid-cell level only, not at the PFT level. Therefore, the PFT-level LAI from CLM is used here, which when integrated over all PFTs is indeed very close to the default total LAI used in GEOS-Chem. Moreover, in the current model configuration the self-healing effect, which counteracts only in part the damaging effect, is already incorporated in the steady-state results.

Referee: Lines 20-24: Is LAI only changing on a monthly basis? If so, is there a large change in LAI from May 31 to June 1 over northern midlatitude forests? How does this influence ozone? Could this impact the authors' results?

The monthly LAI values are all interpolated to obtain daily LAI values that form a more continuous time series in the model.

Referee: Page 10. Line 3: Typo on this line

Corrected.

Referee: Line 30: Is this physically constrained by observations? A decrease of up to 2.6 m2/m2 seems really high. On a similar note, the authors' Figure S3 shows relative changes in LAI due to ozone are pretty high. Could the ozone impact on LAI be overstated?

Long-term responses of canopy-level and grid cell-level LAI to ozone exposure are not well constrained by observations. There is a possibility that these changes are over-stated or understated. That said, the current parameterization uses well-constrained leaf-level statistical information of ozone impacts on photosynthesis and stomatal conductance, and its integrated effect on LAI would be as good or as bad as the current representation of all processes (e.g., allocation) between cellular photosynthesis and leaf growth. CLM-simulated responses of LAI to ozone damage could also be overestimated due to a combination of high biases in simulated LAI for grasses and crops and their demonstrable strong ozone sensitivity. This is now discussed at greater length:

P12 L9: "... This is likely because of the higher sensitivity of grasses and crops to ozone exposure compared with other plant groups in this ozone damage scheme (see Table 1), and the general overestimation of grass and crop LAI in CLM, which is also documented in other studies [Chen et al., 2015; Williams et al., 2016]. The over-representation of grasses and crops in these subtropical and tropical regions in the CLM world may in turn lead to possible high biases in the simulated ozone damage on

LAI therein."

Referee: Page 12. Lines 3-5: Can the authors show this in a figure in supplemental?

We decide that the explanation should be more nuanced in the paragraphs that follow, and thus in this paragraph we eliminate the explanation. The more detailed explanation with figures to support is included in the third paragraph of the section and also in the second last paragraph using the statistical method of Wong et al. [2018]:

P13 L22: "... In the relatively high-NOx regions at northern midlatitudes (over North America, Europe and East Asia), ozone enhancement from ozone-LAI coupling is relatively small compared with the subtropical and tropical regions (Fig. 3b). This is mostly due to the compensation between the effects of reduced dry deposition, which increases ozone, and reduced isoprene emission, which decreases ozone. In Europe and northern China, in particular, ozone-LAI coupling enhances NOx level due to reduced sequestration by biogenic VOCs (Fig. 6b), which further limits ozone production due to more sequestration of OH by NOx (shown in Fig. S8) and thus less efficient cycling of HOx radicals. On the other hand, in the subtropical and tropical regions where isoprene emission is high (Fig. 5a) and NOx level is relatively low (Fig. 6a), reduced dry deposition (Fig. 4b) and reduced isoprene emission (Fig. 5b) add together to enhance ozone concentrations. In central North America, which is low-NOx in general, the small reduction in NOx levels is consistent with the slight ozone reduction there (Fig. 3b)."

Referee: Page 13. Lines 5-9: This sentence is a bit unclear.

We decide that this sentence indeed carries little interesting new insight or scientific significance. We now remove it and join the two paragraphs together. Doing so also creates a smoother narrative overall.

Referee: Page 14. It would be helpful to have the same color scale on Figure 7 and Figure 3b. The Wong et al. (2018) statistical technique needs to be described in

further detail in the manuscript. Does this technique only use the changes in LAI due to ozone damage archived from GEOS-Chem, or does it also use archived ozone concentrations, isoprene and deposition?

The color scale of Fig. 7 is now updated. The statistical method of Wong et al. [2018] is designed to assess the local sensitivity of ozone to any LAI changes and attribute the sensitivity to the two dominant pathways of biogenic emissions and dry deposition. The LAI changes can be caused by any factors, e.g., land use change, $CO_2$ fertilization, etc., and the method is applied here for ozone damage. The description is now modified:

P14 L13: "... using the statistical model developed by Wong et al. [2018], which is a computationally simple, "offline" assessment tool to estimate the local sensitivity of ozone to any LAI changes, whatever the cause of such changes is, and quantify the relative importance of each of the two dominant pathways (dry deposition vs. isoprene emission) in contributing to this sensitivity as a function of an array of variables including mean ozone concentration, total NOx emission, wind speed, temperature, etc., for any vegetated locations. ..."

Referee: Page 15. Line 12: What are the first and second order feedback effects?

Please see our response to the reviewer's question about first-order and second-order effect above.

Referee: Page 17. Line 23: The authors should cite the specific chapter in the IPCC (2013) report that they are referring to, not the entire report.

The reference is now replaced with another one: [Stephens, 2005].

Referee: Line 24: I don't think it is appropriate to describe the formation of ozone from NOx and VOCs "anthropogenic forcing of precursor emissions"

We now modify it to: "anthropogenic precursor emissions". This is more direct and unambiguous.

Referee: Line 20-26: Does this approach assume that the change in ozone due to the LAI feedback + the change in ozone due to NOx emissions = change in ozone due to both NOx emissions + LAI feedback? I think this needs to be stated and the limitations of the approach discussed.

The commutability of the two effects is not a concern for the purpose of quantifying feedback effect. Since LAI changes do not affect anthropogenic NOx emission, NOx emission here is always a prescribed factor and baseline, and the feedback effect of ozone-LAI coupling is examined on top of this baseline by finding the differences between turning on and off the coupling.

Referee: Page 18. Lines 9-10: I'm not sure I understand the difference between "with and without the parameterized ozone-LAI relationship" and "with synchronous vs. asynchronous ozone-LAI coupling". Is the difference that for the first, only GEOS Chem is used, and for the second GEOS Chem and CLM are used? In general, finding a clearer way of describing the different set-ups would be helpful.

Yes, the reviewer's interpretation is correct. We now change the wording to reflect it better:

P19 L6: "... with and without parameterized ozone-LAI coupling within GEOS-Chem (ozone-affected LAI vs. intact, potential LAI), with and without anthropogenic emissions, and with synchronous vs. asynchronous ozone-LAI coupling. ..."

Referee: Line 14: What are the authors defining as long-term here?

It is now specified:

P19 L11: "Generally, ozone damage causes a global LAI reduction for most PFTs under long-term ozone exposure over multiple decades. Compared with ..."

Referee: Line 17: What is the important vegetation structural parameter?

LAI. The sentence now reads: "Only a few studies of ozone-vegetation interactions

have considered this important vegetation structural parameter in coupled model simulations."

Referee: Lines 17-23: I'm a bit confused because the authors say here that the LAI changes are quite different from Sadiq et al. (2017), but then say later on that the "biogeochemical" feedback is similar between the two studies.

Sadiq et al. [2017] considered both biogeochemical and biogeophysical feedbacks. The differences of this study from theirs mainly stem from (decidedly) the exclusion of biogeophysical feedbacks in this study. The biogeochemical part of their study is indeed quite consistent with what we found here.

Referee: Page 19. Lines 2-5: This sentence is a bit unclear. To the best of my understanding, the authors are implying that ozone damage influence on sensible vs. latent heat flux partitioning and the resulting model meteorology is wacky and less certain than the influence of ozone damage on LAI. How do the authors justify this?

The sentence concerned is now removed. The fuller response to the reviewer's concern about the exclusion of biogeophysical feedbacks is also included in our responses above.

Referee: Lines 24-32: I find this particular discussion confusing. Because LAI evolves on slower timescales, I'm not seeing why it is a problem that LAI is updated to reflect ozone damage on the monthly timescale. To me, the bigger issue is whether the authors are resolving seasonal transitions in LAI on monthly or daily timescales. How do the authors know that the parameterization of ozone-LAI relationship is based on the decadal timescale?

The ozone-LAI parameterization is based on multidecadal steady-state results from CLM simulations. The seasonal transitions are handled by an imposed seasonality that is also affected by ozone exposure in the long term. We agree with the reviewer that the wording chosen here may not be the clearest. Since we have now included

more description and explanation of the timescale issue in Sect. 2.5 and Sect. 5 as well (see our responses and cited texts above), we find it appropriate to remove this part altogether from the Conclusions and Discussion section. The sentence regarding "hysteresis effect" is retained, and added to the end of the last paragraph.

Responses to Referee #2

Referee: The main goal of this study is to quantify the impacts of ozone vegetation damage on the atmospheric surface ozone concentrations themselves (ozone air quality). The research team design an intelligent set of systematic global modeling experiments to parse out this particular air quality feedback focused on changes in the LAI only (ignoring meteorology changes). For example, where ozone vegetation damage has been incorporated in coupled chemistry-climate models and Earth system models, (such as NCAR CESM and NASA GISS ModelE2) it is extremely challenging (and maybe impossible) to assess the actual sign and magnitude of this feedback on surface ozone air quality due to the complex bi-directional linkages between the vegetation, meteorology and atmospheric chemistry. The study builds on a recent previous paper: Sadiq, M., Tai, A. P. K., Lombardozzi, D., and Val Martin, M.: Effects of ozone–vegetation coupling on surface ozone air quality via biogeochemical and meteorological feedbacks, Atmos. Chem. Phys., 17, 3055-3066, https://doi.org/10.5194/acp-17-3055-2017, 2017. The previous paper used sensitivity experiments within a single model framework (CLM- CESM) to examine the underlying driving mechanisms for the ozone-vegetation feed- backs. A previous conclusion was that reduced transpiration leading to increased leaf temperatures (and increased BVOC emissions) is an important mechanism in the NH mid-latitudes leading to a fairly strong positive surface ozone response. The previous work is appropriately discussed in the current paper, and a rationale is provided to focus on the relatively smaller feedbacks through LAI changes only in this work.

Referee: The workload represented in this paper is extensive and impressive, including developing an "ozone damage" LAI functional algorithm that is implemented into

GEOS-Chem, and multiple synchronous and asynchronous coupling experiments using CLM and GEOS-Chem. The high quality and clarity of the writing and presentation means that it is possible to follow fairly easily the complex experimental design and methodology. The model results are applied to understand the underlying LAI-related biogeochemical mechanisms (dry deposition versus BVOC emissions only here) that drive the ozone- LAI feedback in the current model framework. The major important new findings of the study are that the O3-LAI feedback can have a different sign on surface ozone AQ depending on region and level of NOx pollution; and that the positive feedback is particularly strong in tropical regions. The study also introduces and calculates a new metric "ozone feedback factor" that is strongly positive in tropical regions, which is an additional important contribution to the literature.

We thank the reviewer for the comments.

Referee: 1. A major finding and possibly the most interesting aspect of the study is the high sensitivity positive feedback in the tropics (through the reduced dry deposition). I believe that this result is based on the application of ozone damage parameters (photosynthesis and stomatal conductance) for temperate plants (e.g. Lombardozzi et al., 2011; 2013)? Is this correct? The entire model framework assumes that tropical plants behave like temperate zone plants in response to ozone? The paper needs to emphasize more strongly that there are essentially no ozone sensitivity measurement data for tropical plants, and therefore the implications for the value of the results.

The Lombardozzi et al. [2013] parameterization scheme used in this study has indeed incorporated results from several tropical plant studies; but the reviewer is right in that the scheme, by pulling data from all studies and sorting them only into three plant groups, effectively treats tropical and temperate plants in the same way. We now address these points and potential problems that arise more fully in the Conclusions and Discussion:

P20 L13: "Uncertainty can arise from the large variability in the ozone sensitivity of

different plants, especially for tropical trees and grasses, which are modeled based on relatively insufficient data as compared with temperature ecosystems [Lombardozzi et al., 2013]. The current Lombardozzi et al. [2015] scheme classifying 15 PFTs into three plant groups is relatively robust in capturing the average plant physiological responses to ozone uptake on a global scale from across many studies, but it treats tropical and temperate plants equivalently as far as ozone sensitivity is concerned, which may lead to possible biases due to an inadequate representation of spatial heterogeneity of plant-ozone ecophysiology. More detailed experimental and field data, especially for tropical and subtropical plants, can potentially help us derive a more region-specific and spatially resolved parameterization that can be particularly useful for high-resolution regional air quality simulations. Along the same line, we find the greatest feedbacks over tropical forests, where ozone concentrations and fluxes are not as well constrained by available in-situ observations as in the midlatitudes. More extensive and long-term measurements of biosphere-atmosphere fluxes in tropical regions are necessary to ascertain the strength of ozone-vegetation feedbacks in these identified hotspots. We also note that our parameterization necessarily ignores hysteresis effect, whereby damage done at incidentally high ozone concentrations may not undergo full recovery even when ozone levels drop again."

Referee: 2. The exponential LAI parameterization function for GEOS-Chem. Based on Figure 1, the saturation occurs for relatively low ozone. Between 40 and 100 ppbv there is no dependence of LAI on ozone concentration. This function seems to be physically unrealistic. We would expect the LAI response of a forest or cropland ecosystem growing in âĹij45 ppbv ambient ozone to be rather different to such in âĹij90 ppbv ambient ozone?

The relationship between LAI and ozone exposure is poorly constrained by observations, since there have been no long-term monitoring or experimental studies to investigate the responses of LAI to ozone damage at a wide enough range of different ozone levels and for a sufficient diversity of PFTs. Most experimental studies to date

only considered ambient vs. elevated ozone without finer gradation of ozone levels, so no continuous functional relationship can be inferred from their results.

The currently simulated relationship between LAI and ozone is consistent with the ozone damage function for photosynthesis and stomatal conductance [Lombordozzi et al., 2013] used in this study, whereby most of the ozone effect on photosynthesis and conductance happens when cumulative ozone uptake (CUO) passes a certain threshold, beyond which the correlations between CUO and damage are weak. The damage function was developed based on the most comprehensive data available to date, which showed that there is substantial variability in plant responses to ozone and often the photosynthetic and stomatal responses are not correlated with CUO. Even within one species across different studies with the same experimental conditions, correlations are not evident. Correlations of photosynthesis or stomatal conductance with CUO are often only evident within a given study that measures the response of the same plant through time, but using the results of individual studies as the basis for universal parameterization may not be a valid representation of global ecosystems, and thus we consider it more robust to use a function based on responses across multiple species.

Moreover, the response of LAI to ozone is the outcome of the integrated effect of ozone damage on photosynthesis and stomatal conductance, and all subsequent processes (e.g., allocation, growth/senescence). In the model, what appears to be happening is a self-protective effect, namely, that ozone-induced stomatal closure limits further ozone uptake and damage so that LAI does not decline infinitely as ambient ozone increases further. This is explained in Sect. 2.4, but an empirical basis for such a mechanism is lacking from the literature.

Referee: 3. Regarding the LAI function. The paper could be greatly strengthened by showing validation and evaluation of the LAI function against measurement data (or even plant biomass could be used as proxy for some ecosystems where LAI changes are less available).

We agree with the reviewer that a validation and evaluation with empirical data will strengthen the implications of our study. As mentioned above, however, there have been no experimental studies to investigate the responses of LAI to ozone damage at a graduation of ozone levels that is sufficient for inferring any continuous functional formulation between ozone levels and structural variables such as LAI and biomass. We can alternatively at least compare our simulated reduction in LAI at some prescribed ambient vs. elevated ozone levels with studies that measured or meta-analyzed vegetation changes (in LAI, biomass, yield, etc.) at these ozone levels [e.g., Karnosky et al., 2005; Mills et al., 2007; Dermody et al., 2008; Feng et al., 2008]. The percentage reductions predicted by our parameterization typically fall well within their empirical uncertainty bounds. We now include this comparison in Sect. 2.4:

P9 L9: "... When we apply some generic ambient ozone level (e.g., 30 ppb) and an elevated (+50%) ozone level to Eq. (8), the percentage changes in LAI as ozone increases from ambient to elevated level range between about –20% and +3% (with the bottom and top 2.5% of grid cells/PFTs trimmed). These modeled LAI changes generally fall within the empirical uncertainty bounds found by previous ozone-elevation experiments for a few species of trees and crops [e.g., Karnosky et al., 2005; Dermody et al., 2008; Feng et al., 2008]."

Referee: 4. The paper assumes that BVOC emissions are essentially positive linear function of LAI. In reality, ozone vegetation damage may influence BVOC emissions in complex ways (even independent of LAI) through changes in biochemistry and plant production, and even lead to increases in BVOC emissions. There is a growing literature in this area that needs to be cited and discussed. The paper needs to emphasize the limitations of the BVOC modeling response and that the BVOC response sign could be different on monthly timescales (positive versus negative).

The complex interactions between ozone exposure and terpenoid biosynthesis are now discussed in the Conclusions and Discussion:

P20 L28: "... This study also considers ozone effects on biogenic VOC emissions only via the gradual modification of LAI, but previous studies have suggested that chronic ozone exposure may inhibit isoprene biosynthesis by directly interfering with enzymatic activities [e.g., Calfapietra et al., 2007], and high ozone episodes may even enhance isoprene emission by triggering plant defense mechanisms against oxidative stress [e.g., Fares et al., 2006]. It is necessary to further examine the interactions between ozone and isoprene biosynthesis on different timescales. Moreover, in this study meteorological conditions are prescribed and looped over for a typical year only, and thus the potential impacts of interannual climate variability on the ozone-LAI relationships are not fully considered. For instance, the occurrence of droughts may either weaken or strengthen the coupling between plants and ozone by interfering with photosynthetic capacity, stomatal behaviors and biogenic emissions [e.g., Wang et al., 2017]. Despite the limitations stated above, our findings still attest to the existence of strong ozone biogeochemical feedbacks under typical conditions, ..."

Referee: 5. The study seems to only consider isoprene? I agree isoprene is by far the most important for ozone, but how do changes in other BVOCs influence the ozone-LAI feedbacks? For example, monoterpenes and sesquiterpenes in the tropics? Both CLM and GEOS-Chem do include higher level BVOCs and terpenes.

The reviewer is right in that isoprene, among all BVOCs, is by far the most important for ozone-vegetation interactions. This is indeed the common finding from our other studies using both GEOS-Chem and CESM, both of which consider dynamic BVOC emissions and chemistry in addition to isoprene, e.g., Sadiq et al. [2017] and Wong et al. [2018]. They typically found that other BVOCs play only a very small role in ozone-vegetation feedbacks.

Referee: 6. There are some curious features of Figure 3a, the baseline surface ozone distribution, in this GEOS-Chem model. For example, (i) surface ozone in eastern China is about 40-50 ppbv, much lower (about half of the levels) than in Northeastern US (70-80 ppbv). Indian subcontinent has very low ozone whereas Sahara Desert

has substantially higher ozone. The highest European values in summer are over the Mediterranean Sea rather than the continental land mass. Do these features agree with current ozone measurement monitoring networks in these regions?

GEOS-Chem reproduces observed ozone concentrations to various extents of success depending on the region. For instance, while it is possible that ozone in eastern China is lower than in the eastern US due to weaker solar radiation reaching the surface in China, the problem of positive biases in simulated surface ozone in the eastern US is also well documented. For other regions where only satellite-derived ozone columns are available for model validation, GEOS-Chem also has varying degrees of success in matching observations. More about GEOS-Chem performance is now included in Sect. 2.3:

P6 L22: "... In general, GEOS-Chem underestimates tropospheric ozone in the tropics but overestimates it in the northern subtropics and southern midlatitudes [Zhang et al., 2010]. For regional surface ozone, the model has small systematic biases overall in the US and China, but has a tendency to overestimate summertime concentrations in the eastern US and certain sites in China [Wang et al., 2009; Wang et al., 2011; Zhang et al., 2011]. Anthropogenic emissions ..."

Referee: 7. Related to Figure 3b, the authors offer an explanation for the decreased ozone signals in US crop belt and North China Plain: "Such a reduction is driven by reduced transport of VOCs as well as organic nitrate formed from VOC-NOx reactions following reduced LAI elsewhere in more vegetated regions". For sure, their model shows limited to no LAI changes in these regions. However, it is really interesting that these regions are heavily dominated by crop ecosystems where we would expect to see substantial relative changes in LAI due to ozone damage in reality. Are specific crop types represented in the CLM model version? Would the results be different in sign if specific crop types are represented in the model?

The sentence concerned has already been removed in response to the other reviewer's

similar question. Indeed, it is likely that local LAI changes in central North America may still play a role in the negative ozone changes there, as is shown in Fig. 7. See our response above.

Regarding crop types, the CLM land cover we used (and thus in GEOS-Chem) only considers C3 crops generically without resolving specific crop types. Since the differences in ozone sensitivity among different crop types are typically less than that between crops and other vegetation types, we deem it unlikely that differentiating among different crop types would lead to substantially different results.

Referee: 8. The difference between the synchronous and asynchronous coupling methods (Figure 3b versus 8b) are massive for the [O3] changes due to O3-LAI coupling. At least the sign is the same, but the spatial responses are very different, especially over N America, Europe, Central Asia, Middle East, N Africa and E China. In many of these widespread regions, synchronous shows a strong signal, but asynchronous has no signal. The authors state: "Most of the bigger differences occur in low-LAI regions which are more prone to idiosyncratic model (CLM) behaviors and numerical outliers especially in the asynchronously coupled cases where such peculiarities are not smoothed out." The paper needs to offer a more scientific, and more physically mechanistic explanation for these differences (rather than "idiosyncratic model (CLM) behaviors"). What does "smoothed out" mean? How can readers know which is the most realistic response? Please directly link the results to the 3 reasons for doing the asynchronous experiments (Page 15, Lines 10-15).

A more detailed explanation for the discrepancies in low-LAI regions is now included.

P16 L18: "... The overall strong resemblance in the relative LAI changes between the two coupling approaches, at least for regions with sufficiently high LAI, suggests that the simplified parameterization for ozone-LAI coupling on a monthly timescale used in synchronous coupling (Sect. 2.4) is a reasonable idealization of the cumulative long-term steady-state responses of vegetation to temporally varying ozone levels."

P16 L29: "... Most of the bigger differences occur in low-LAI regions and over the oceans. The discrepancies likely arise from the tendency toward more unstable model (CLM) behaviors at low LAI and the inclusion of diurnally and daily fluctuating ozone in ozone-LAI coupling in the asynchronous approach (as opposed to using constant ozone levels in parameterizing the ozone-LAI relationship). In the original ozone-vegetation scheme in CLM, cumulative ozone damage only occurs when both LAI and ozone level are above some thresholds. Thus, when LAI is low and ozone fluctuating, ozone-LAI coupling becomes more erratic and loses persistence. Such peculiarities are smoothed out when parameterizing the ozone-LAI relationship by the best-fitting of responses curves and filtering of poorly fitting locations."

Referee: Minor comments Fig. 7 Title "Attribution" Corrected.

References: Dermody, O., Long, S. P., McConnaughay, K., and DeLucia, E. H.: How do elevated CO2 and O3 affect the interception and utilization of radiation by a soybean canopy?, Global Change Biol., 14, 556-564, 10.1111/j.1365-2486.2007.01502.x, 2008. Feng, Z. Z., Kobayashi, K., and Ainsworth, E. A.: Impact of elevated ozone concentration on growth, physiology, and yield of wheat (triticum aestivum l.): A meta-analysis, Global Change Biol., 14, 2696-2708, 10.1111/j.1365-2486.2008.01673.x, 2008. Karnosky, D. F., Pregitzer, K. S., Zak, D. R., Kubiske, M. E., Hendrey, G. R., Weinstein, D., Nosal, M., and Percy, K. E.: Scaling ozone responses of forest trees to the ecosystem level in a changing climate, Plant Cell Environ., 28, 965-981, 2005. Mills, G., Buse, A., Gimeno, B., Bermejo, V., Holland, M., Emberson, L., and Pleijel, H.: A synthesis of aot40-based response functions and critical levels of ozone for agricultural and horticultural crops, Atmos. Environ., 41, 2630-2643, doi:10.1016/J.Atmosenv.2006.11.016, 2007.

Please also note the supplement to this comment:
https://www.atmos-chem-phys-discuss.net/acp-2018-351/acp-2018-351-AC1-supplement.pdf